# On kernel-based statistical learning in the mean field limit

**Christian Fiedler**[1]    **Michael Herty**[2]    **Sebastian Trimpe**[3]

[1,3]Institute for Data Science in Mechanical Engineering (DSME)
RWTH Aachen University, Aachen, Germany
`{fiedler,trimpe}@dsme.rwth-aachen.de`

[2] Institute for Geometry and Practical Mathematics (IGPM)
RWTH Aachen University, Aachen, Germany
`herty@igpm.rwth-aachen.de`

## Abstract

In many applications of machine learning, a large number of variables are considered. Motivated by machine learning of interacting particle systems, we consider the situation when the number of input variables goes to infinity. First, we continue the recent investigation of the mean field limit of kernels and their reproducing kernel Hilbert spaces, completing the existing theory. Next, we provide results relevant for approximation with such kernels in the mean field limit, including a representer theorem. Finally, we use these kernels in the context of statistical learning in the mean field limit, focusing on Support Vector Machines. In particular, we show mean field convergence of empirical and infinite-sample solutions as well as the convergence of the corresponding risks. On the one hand, our results establish rigorous mean field limits in the context of kernel methods, providing new theoretical tools and insights for large-scale problems. On the other hand, our setting corresponds to a new form of limit of learning problems, which seems to have not been investigated yet in the statistical learning theory literature.

## 1 Introduction

Models with many variables play an important role in many fields of mathematical and physical sciences. In this context, going to the limit of infinitely many variables is an important analysis and modeling approach. Interacting particle systems are a classic example; these are usually modeled as dynamical systems describing the temporal evolution of many interacting objects. In physics, such systems were first investigated in the context of gas dynamics, cf. [1]. Since even small volumes of gases typically contain an enormous number of molecules, a microscopic modeling approach quickly becomes infeasible and one considers the evolution of densities instead [2]. In the past decades, interacting particle systems arising from many different domains have been considered, for example, animal movement (inter alia swarms of birds, schools of fish, colonies of microorganisms) [3, 4], social and political dynamics [5, 6], crowd modeling and control (pedestrian movement, gathering at large events like football games or concerts) [7–9], swarms of robots [10–12] or vehicular traffic (in particular, traffic jams) [13]. There is now a vast literature on such applications, and we refer to the surveys [14–16] as starting points. A prototypical example of such a system is given by $\dot{x}_i = \frac{1}{M} \sum_{j=1}^{M} \phi(x_i, x_j)(x_j - x_i)$, for $i = 1, \ldots, M$, where $M \in \mathbb{N}_+$ particles or agents are modelled by their state $x_i \in \mathbb{R}^d$, $i = 1, \ldots, M$, evolving according to some interaction rule $\phi : \mathbb{R}^d \times \mathbb{R}^d \to \mathbb{R}$. Typical questions then concern the long-term behavior of such systems, in particular, emergent phenomena like consensus or alignment [17]. While first-principles modeling

37th Conference on Neural Information Processing Systems (NeurIPS 2023).

has been very successful for interacting particle systems in physical domains, using this approach to model the interaction rules in complex domains like social and opinion dynamics, pedestrian and animal movement or vehicular traffic, can be problematic. Therefore, learning interaction rules from data has been recently intensively investigated, for example, in the pioneering works [18, 19]. The data consists typically of (sampled) trajectories of the particle states, potentially with measurement noise, and the goal is to learn a good approximation of the interaction rule $\phi$.

Our work is motivated by a related problem. Frequently, the state of such a complex multiagent system can be easily measured or estimated, e.g., by video recordings or image snapshots for bird swarms or schools of fish, and microscopy recordings for microorganism colonies; aerial imaging for human crowds (e.g., via quadcopters); and polling and social media analysis for opinion dynamics. However, some interesting features of the whole system might be more difficult to measure. For example, how a swarm of birds or a school of fish will react to an external stimulus (like an approaching predator), given the current state of the population. Such a reaction could be a change of density or spread of the population, or a change in mean velocity. Another example is given by features of a society in opinion dynamics (average happiness, aggression potential, susceptibility to adversarial interventions), given the current "opinion state". Measuring such features can be difficult, for example, due to a required intervention. Formally, such a feature is a functional $F_M : (\mathbb{R}^d)^M \to \mathbb{R}$ of the current state of the system, and since the state is often easy to measure, it would be useful to have an explicit mapping from state to feature of interest. However, since first principles modeling is unlikely to be successful in the domains considered here, it is promising to learn such a mapping from data. We can formalize this as a standard supervised learning task: The data set consists of $D_N^{[M]} = ((\vec{x}_1, y_1), \ldots, (\vec{x}_N, y_N))$, where $\vec{x}_n \in (\mathbb{R}^d)^M$ are snapshot measurements of the particle states (corresponding to the input of the functional) and $y_n \in \mathbb{R}$ is the value of the functional of interest, potentially with measurement noise, at snapshot state $\vec{x}_n$. Let us assume an additive noise model, i.e., $y_n = F_M(\vec{x}_n) + \epsilon_n$ for $n = 1, \ldots, N$, where $\epsilon_1, \ldots, \epsilon_N \in \mathbb{R}$ are noise variables. This is now a regression problem that could be solved, for example, using a Support Vector Machine (SVM) [20]. Note that for this we need a kernel $k_M : (\mathbb{R}^d)^M \times (\mathbb{R}^d)^M \to \mathbb{R}$ on $(\mathbb{R}^d)^M$.

Similarly to classical physical examples like gas dynamics, the case of a large number of particles is also relevant in modern complex interacting particle systems. Since this poses computational and modeling challenges, it can be advantageous to go also here to a kinetic level and model the evolution of the particle distribution instead of every individual particle. It is well-established how to derive a kinetic partial differential equation from ordinary differential equations systems on the particle level, for example, using the Boltzmann equation or via a mean field limit, cf. [17] for an overview in the context of multi-agent systems. Formally, instead of trajectories of particle states of the form $[0, T] \ni t \mapsto \vec{x}(t) \in (\mathbb{R}^d)^M$, we then have trajectories of probability measures $[0, T] \ni t \mapsto \mu(t) \in \mathcal{P}(\mathbb{R}^d)$. This immediately raises the question of whether the learning setup outlined above also allows a corresponding kinetic limit. More precisely, let $K \subseteq \mathbb{R}^d$ be compact and assume that all particles remain confined to this compactum, i.e., $x_i(t) \in K$ for all $i = 1, \ldots, M$ and all $t \in [0, T]$ under the microscopic dynamics.[1] If the underlying dynamics have a mean field limit, then it is reasonable to assume that the finite-input functionals $F_M : K^M \to \mathbb{R}$ converge also in mean field to some $F : \mathcal{P}(K) \to \mathbb{R}$ for $M \to \infty$, see Section 2 for a precise definition of this notion. In turn, we can now formulate a corresponding learning problem on the mean field level: A data set is then given by $D_N = ((\mu_1, y_1), \ldots, (\mu_N, y_N))$, where $\mu_n \in \mathcal{P}(K)$ are snapshots of the particle state distribution over time and $y_n \in \mathbb{R}$ are again potentially noisy measurements of the functional. Assuming an additive noise model, this corresponds to $y_n = F(\mu_n) + \epsilon_n$, $n = 1, \ldots, N$. If we want to use an SVM on the kinetic level, we need a kernel $k : \mathcal{P}(K) \times \mathcal{P}(K) \to \mathbb{R}$ on probability distributions. There are several options available for this, see e.g. [21]. However, assuming that all ingredients of the learning problem arise as a mean field limit, this naturally leads to the question of whether a mean field limit of kernels exists, and what this means for the relation of the learning problems on the finite-input and kinetic level. In [22], this reasoning has motivated the introduction and investigation of the mean field limit of kernels. In the present work, we extend the theory of these kernels and investigate them in the context of statistical learning theory. In particular, since in practice one would use the mean field kernels on microscopic data with large, but finite $M$, we need convergence results of the various objects appearing in statistical learning with kernels. Exactly such results are provided in Section 4.

---

[1]This means the dynamics on the level of individual particles.

Finally, we would like to stress that the technical developments here are independent of the motivation outlined above, in that they apply to mean field limits of functions and kernels that do not necessarily arise form the dynamics of interacting particle systems.

**Contributions** Our contributions cover three closely related aspects. 1) We extend and complete the theory of mean field limit kernels and their RKHSs (Section 2). In Theorem 2.3, we precisely describe the relationship between the RKHS of the finite-input kernels and the RKHS of the mean field kernel, completing the results from [22]. In particular, this allows us to interpret the latter RKHS as the mean field limit of the former RKHSs. Furthermore, in Lemma 2.4 and 2.5, we provide inequalities for the corresponding RKHS norms, which are necessary for $\Gamma$-convergence arguments. 2) We provide results relevant for approximation with mean field limit kernels (Section 3). With Proposition 3.1, we give a first result on the approximation power of mean field limit kernels, and in Theorem 3.3 we can also provide a representer theorem for these kernels. For its proof, we use a $\Gamma$-convergence argument, which is to the best of our knowledge the first time this technique has been used in the context of kernel methods. 3) We investigate the mean field limit of kernels in the context of statistical learning theory (Section 4). We first establish an appropriate mean field limit setup for statistical learning problems, based on a slightly stronger mean field limit existence result than available so far, cf. Proposition 2.1. To the best of our knowledge, this is a new form of a limit for learning problems. In this setup, we then provide existence, uniqueness, and representer theorems for empirical and (using an apparently new notion of mean field convergence of probability distributions) infinite-sample solutions of SVMs, cf. Proposition 4.3 and 4.5. Finally, under a uniformity assumption, we can also establish convergence of the minimal risks in Proposition 4.7.

Our developments are relevant from two different perspectives: on the one hand, they constitute a theoretical proof-of-concept that the mean field limit can be "pulled through" the (kernel-based) statistical learning theory setup. In particular, this demonstrates that rigorous theoretical results can be transferred through the mean field limit, similar to works in the context of control of interacting particle systems, see e.g. [23]. On the other hand, our setup appears to be a new variant of a large-number-of-variables limit in the context of machine learning, complementing established settings like infinite-width neural networks [24].

Due to space constraints, all proofs and some additional technical results have been placed in the supplementary material.

## 2 Kernels and their RKHSs in the mean field limit

**Setup and preliminaries** Let $(X, d_X)$ be a compact metric space and denote by $\mathcal{P}(X)$ the set of Borel probability measures on $X$. We endow $\mathcal{P}(X)$ with the topology of weak convergence of probability measures. Recall that for $\mu_n, \mu \in \mathcal{P}(X)$, we say that $\mu_n \to \mu$ weakly if for all bounded and continuous $f : X \to \mathbb{R}$ (since $X$ is compact, this is equivalent to $f$ continuous) we have $\lim_{n \to \infty} \int_X \phi(x) \mathrm{d}\mu_n(x) \to \int_X \phi(x) \mathrm{d}\mu(x)$. The topology of weak convergence can be metrized by the Kantorowich-Rubinstein metric $d_{\mathrm{KR}}$, defined by

$$ d_{\mathrm{KR}}(\mu_1, \mu_2) = \sup \left\{ \int_X \phi(x) \mathrm{d}(\mu_1 - \mu_2)(x) \mid \phi : X \to \mathbb{R} \text{ is 1-Lipschitz} \right\}. $$

Note that since $X$ is compact and hence separable, the Kantorowich-Rubinstein metric is equal to the 1-Wasserstein metric here. Furthermore, $\mathcal{P}(X)$ is compact in this topology. For $M \in \mathbb{N}_+$ and $\vec{x} \in X^M$, denote the $i$-th component of $\vec{x}$ by $x_i$, and define the *empirical measure* for $\vec{x}$ by $\hat{\mu}[\vec{x}] = \frac{1}{M} \sum_{i=1}^{M} \delta_{x_i}$, where $\delta_x$ denotes the Dirac measure centered at $x \in X$. The empirical measures are dense in $\mathcal{P}(X)$ w.r.t. the Kantorowich-Rubinstein metric. Additionally, define $d_{\mathrm{KR}}^2 : \mathcal{P}(X)^2 \times \mathcal{P}(X)^2 \to \mathbb{R}_{\geq 0}$ by $d_{\mathrm{KR}}^2((\mu_1, \mu_1'), (\mu_2, \mu_2')) = d_{\mathrm{KR}}(\mu_1, \mu_2) + d_{\mathrm{KR}}(\mu_1', \mu_2')$, and note that $(\mathcal{P}(X)^2, d_{\mathrm{KR}}^2)$ is a compact metric space. Moreover, denote the set of permutations on $\{1, \ldots, M\}$ by $\mathcal{S}_M$, and for a tuple $\vec{x} \in X^M$ and permutation $\sigma \in \mathcal{S}_M$ define $\sigma\vec{x} = (x_{\sigma(1)}, \ldots, x_{\sigma(M)})$. Finally, we recall some well-known definitions and results from the theory of reproducing kernel Hilbert spaces, following [20, Chapter 4]. For an arbitrary set $\mathcal{X} \neq \emptyset$ and a Hilbert space $(H, \langle \cdot, \cdot \rangle_H)$ of functions on $\mathcal{X}$, we say that a map $k : \mathcal{X} \times \mathcal{X} \to \mathbb{R}$ is a *reproducing kernel* for $H$ if 1) $k(\cdot, x) \in H$ for all $x \in \mathcal{X}$; 2) for all $x \in \mathcal{X}$ and $f \in H$ we have $f(x) = \langle f, k(\cdot, x) \rangle_H$. Note that if a reproducing kernel exists, it is unique. If such a Hilbert space has a reproducing kernel, we call $H$ a reproducing kernel Hilbert space (RKHS) and $k$ its (reproducing) kernel. It is well-known that a reproducing kernel is

symmetric and positive semidefinite, and that every symmetric and positive semidefinite function has a unique RKHS for which it is the reproducing kernel. For brevity, if $k$ is symmetric and positive semidefinite, or equivalently, if it is the reproducing kernel of an RKHS, we call $k$ simply a kernel, and denote by $(H_k, \langle \cdot, \cdot \rangle_k)$ its unique associated RKHS. Define also $H_k^{\text{pre}} = \text{span}\{k(\cdot, x) \mid x \in \mathcal{X}\}$, then for $f = \sum_{n=1}^{N} \alpha_n k(\cdot, x_n) \in H_k^{\text{pre}}$ and $g = \sum_{m=1}^{M} \beta_m k(\cdot, y_m) \in H_k^{\text{pre}}$ we have $\langle f, g \rangle_k = \sum_{n=1}^{N} \sum_{m=1}^{M} \alpha_n \beta_m k(y_m, x_n)$, and $H_k^{\text{pre}}$ is dense in $H_k$.

**The mean field limit of functions and kernels**   Given $f_M : X^M \to \mathbb{R}$, $M \in \mathbb{N}_+$, and $f : \mathcal{P}(X) \to \mathbb{R}$, we say that $f_M$ *converges in mean field to* $f$ and that $f$ is the (or a) *mean field limit* of $f_M$, if $\lim_{M \to \infty} \sup_{\vec{x} \in X^M} |f_M(\vec{x}) - f(\hat{\mu}[\vec{x}])| = 0$. In this case, we write $f_M \xrightarrow{\mathcal{P}_1} f$. Let now $(Y, d_Y)$ be another metric space and $f_M : X^M \times Y \to \mathbb{R}$, $M \in \mathbb{N}_+$, and $f : \mathcal{P}(X) \times Y \to \mathbb{R}$, then we say that $f_M$ *converges in mean field to* $f$ and that $f$ is the (or a) *mean field limit* of $f_M$, if for all compact $K \subseteq Y$ we have

$$\lim_{M \to \infty} \sup_{\vec{x} \in X^M, y \in K} |f_M(\vec{x}, y) - f(\hat{\mu}[\vec{x}], y)| = 0. \tag{1}$$

and also write $f_M \xrightarrow{\mathcal{P}_1} f$. The following existence results for mean field limits is slightly more general than what is available in the literature, and it is essentially a direct generalization of [25, Theorem 2.1], in the form of [26, Lemma 1.2].

**Proposition 2.1.** Let $(X, d_X)$ be a compact metric space and $(Z, d_Z)$ a metric space that has a countable basis $(U_n)_n$ such that $\bar{U}_n$ is compact for all $n \in \mathbb{N}$. Let $f_M : X^M \times Z \to \mathbb{R}$, $M \in \mathbb{N}_+$, be a sequence of functions fulfilling the following conditions: 1) *(Symmetry in $\vec{x}$)*[2] For all $M \in \mathbb{N}_+$, $\vec{x} \in X^M$, $z \in Z$ and permutations $\sigma \in \mathcal{S}_M$, we have $f_M(\sigma \vec{x}, z) = f_M(\vec{x}, z)$; 2) *(Uniform boundedness)* There exists $B_f \in \mathbb{R}_{\geq 0}$ and a function $b : Z \to \mathbb{R}_{\geq 0}$ such that $\forall M \in \mathbb{N}_+, \vec{x} \in X^M, z \in z : |f_M(\vec{x}, z)| \leq B_f + b(z)$; 3) *(Uniform Lipschitz continuity)* There exists some $L_f \in \mathbb{R}_{>0}$ such that for all $M \in \mathbb{N}_+$, $\vec{x}_1, \vec{x}_2 \in X^M$, $z_1, z_2 \in Z$ we have $|f_M(\vec{x}_1, z_1) - f_M(\vec{x}_2, z_2)| \leq L_f (d_{\text{KR}}(\hat{\mu}[\vec{x}_1], \hat{\mu}[\vec{x}_2]) + d_Z(z_1, z_2))$.

Then there exists a subsequence $(f_{M_\ell})_\ell$ and a continuous function $f : \mathcal{P}(X) \times Z \to \mathbb{R}$ such that $f_{M_\ell} \xrightarrow{\mathcal{P}_1} f$ for $\ell \to \infty$. Furthermore, $f$ is $L_f$-Lipschitz continuous and there exists $B_F \in \mathbb{R}_{\geq 0}$ such that for all $\mu \in \mathcal{P}(X)$, $z \in Z$ we have $|f(\mu, z)| \leq B_F + b(z)$.

We now turn to the mean field limit of kernels as introduced in [22]: Given $k_M : X^M \times X^M \to \mathbb{R}$ and $k : \mathcal{P}(X) \times \mathcal{P}(X) \to \mathbb{R}$, we say that $k_M$ *converges in mean field to* $k$ and that $k$ is the (or a) *mean field limit* of $k_M$, if

$$\lim_{M \to \infty} \sup_{\vec{x}, \vec{x}' \in X^M} |k_M(\vec{x}, \vec{x}') - k(\hat{\mu}[\vec{x}], \hat{\mu}[\vec{x}'])| = 0. \tag{2}$$

In this case we write $k_M \xrightarrow{\mathcal{P}_1} k$.

For convenience, we recall [22, Theorem 2.1], which ensures the existence of a mean field limit of a sequence of kernels.

**Proposition 2.2.** Let $k_M : X^M \times X^M \to \mathbb{R}$ be a sequence of kernels fulfilling the following conditions. 1) *(Symmetry in $\vec{x}$)* For all $M \in \mathbb{N}_+$, $\vec{x}, \vec{x}' \in X^M$ and permutations $\sigma \in \mathcal{S}_M$ we have $k_M(\sigma \vec{x}, \vec{x}') = k_M(\vec{x}, \vec{x}')$; 2) *(Uniform boundedness)* There exists $C_k \in \mathbb{R}_{\geq 0}$ such that $\forall M \in \mathbb{N}_+, \vec{x}, \vec{x}' \in X^M : |k_M(\vec{x}, \vec{x}')| \leq C_k$; 3) *(Uniform Lipschitz continuity)* There exists some $L_k \in \mathbb{R}_{>0}$ such that for all $M \in \mathbb{N}_+$, $\vec{x}_1, \vec{x}_1', \vec{x}_2, \vec{x}_2' \in X^M$ we have $|k_M(\vec{x}_1, \vec{x}_1') - k_M(\vec{x}_2, \vec{x}_2')| \leq L_k d_{\text{KR}}^2 [(\hat{\mu}[\vec{x}_1], \hat{\mu}[\vec{x}_1']), (\hat{\mu}[\vec{x}_2], \hat{\mu}[\vec{x}_2'])]$.

Then there exists a subsequence $(k_{M_\ell})_\ell$ and a continuous kernel $k : \mathcal{P}(X) \times \mathcal{P}(X) \to \mathbb{R}$ such that $k_{M_\ell} \xrightarrow{\mathcal{P}_1} k$, and $k$ is also bounded by $C_k$.

Let $k_M : X^M \times X^M \to \mathbb{R}$ be a given sequence of kernels fulfilling the conditions of Proposition 2.2. Then there exists a subsequence $(k_{M_\ell})_\ell$ converging in mean field to a kernel $k : \mathcal{P}(X) \times \mathcal{P}(X) \to \mathbb{R}$.

---

[2]As is well-known, cf. [26, Remark 1.1.3], this condition is actually implied by the next condition. However, as usual in the kinetic theory literature, we kept this condition for emphasis.

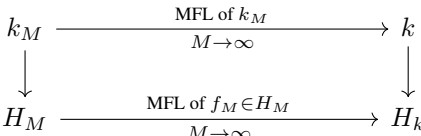

Figure 1: The kernel $k$ arises as the mean field limit (MFL) of the kernels $k_M$ (Proposition 2.2). Every uniformly norm-bounded sequence $f_M \in H_M$, $M \in \mathbb{N}_+$, has an MFL in $H_k$, and every function $f \in H_k$ arises as such an MFL (Theorem 2.3). Based on [22, Figure 1].

*From now on, we only consider this subsequence* and denote it again by $(k_M)_M$, i.e., $k_M \xrightarrow{\mathcal{P}_1} k$. Unless noted otherwise, every time we need a further subsequence, we will make this explicit.[3]

**The RKHS of the mean field limit kernel**  Denote by $H_M := H_{k_M}$ the (unique) RKHS corresponding to kernel $k_M$ and denote by $H_k$ the unique RKHS of $k$. For basic properties of these objects as well as classes of suitable kernels we refer to [22].

We clarify the relation between $H_M$ and $H_k$ in the next result.

**Theorem 2.3.** 1) For every $f \in H_k$, there exists a sequence $f_M \in H_M$, $M \in \mathbb{N}_+$, such that $f_M \xrightarrow{\mathcal{P}_1} f$. 2) Let $f_M \in H_M$ be sequence such that there exists $B \in \mathbb{R}_{\geq 0}$ with $\|f_M\|_M \leq B$ for all $M \in \mathbb{N}_+$. Then there exists a subsequence $(f_{M_\ell})_\ell$ and $f \in H_k$ with $f_{M_\ell} \xrightarrow{\mathcal{P}_1} f$ and $\|f\|_k \leq B$.

In other words, on the one hand, every RKHS function from $H_k$ arises as a mean field limit of RKHS functions from $H_M$. On the other hand, every uniformly norm-bounded sequence of RKHS functions $(f_M)_M$ has a mean field limit in $H_k$.

Note that the preceding result is considerably stronger than the corresponding results in [22]: In contrast to [22, Theorem 4.4] we do not need to go to another subsequence in the first item, and we ensure that the mean field limit $f$ is contained in $H_k$ (and norm-bounded by the same uniform bound), which was missing from Corollary 4.3 in the same reference.

The relation between the kernels $k_M$ and their RKHSs $H_M$, and the mean field limit kernel $k$ and its RKHS $H_k$ is illustrated as a commutative diagram in Figure 1. In order to arrive at the mean field RKHS $H_k$, on the one hand, we consider the mean field limit $k$ of the $k_M$, and then form the corresponding RKHS $H_k$. This is essentially the content of Proposition 2.2. On the other hand, we can first go from the kernel $k_M$ to the associated unique RKHS $H_M$ (for each $M \in \mathbb{N}_+$). Theorem 2.3 then says that $H_k$ can be interpreted as a mean field limit of the RKHSs $H_M$, since every function in $H_k$ arises as a mean field limit of a sequence of functions from the $H_M$, and every uniformly norm-bounded sequence of such functions has a mean field limit that is in $H_k$.

Next, we state two technical results that will play an important role in the following developments, and which might be of independent interest. They describe $\liminf$ and $\limsup$ inequalities required for $\Gamma$-convergence arguments used later on.

**Lemma 2.4.** Let $f_M \in H_M$, $M \in \mathbb{N}_+$, and $f \in H_k$ such that $f_M \xrightarrow{\mathcal{P}_1} f$, then

$$\|f\|_k \leq \liminf_{M \to \infty} \|f_M\|_M. \tag{3}$$

**Lemma 2.5.** Let $f \in H_k$. Then there exist $f_M \in H_M$, $M \in \mathbb{N}_+$, such that $\lim_{M \to \infty} \sup_{\vec{x} \in X^M} |f_M(\vec{x}) - f(\hat{\mu}[\vec{x}])| = 0$, and

$$\limsup_{M \to \infty} \|f_M\|_M \leq \|f\|_k. \tag{4}$$

## 3  Approximation with kernels in the mean field limit

Kernel-based machine learning methods use in general an RKHS as the hypothesis space, and learning often reduces to a search or optimization problem over this function space. For this reason, it is

---

[3]It is customary in the kinetic theory literature to switch to such a subsequence. However, for some results that are about to follow, it is important that no further switch to a subsequence happens, hence we need to be more explicit in these cases.

important to investigate the approximation properties of a given kernel and its associated RKHS as well as to ensure that the learning problem over an RKHS (which is in general an infinite-dimensional object) can be tackled with finite computations.

The next result asserts that, under a uniformity condition, the approximation power of the finite-input kernels $k_M$ is inherited by the mean field limit kernel.

**Proposition 3.1.** For $M \in \mathbb{N}_+$, let $\mathcal{F}_M$ be the set of symmetric functions that are continuous w.r.t. $(\vec{x}, \vec{x}') \mapsto d_{\mathrm{KR}}(\hat{\mu}[\vec{x}], \hat{\mu}[\vec{x}'])$. Let $\mathcal{F} \subseteq C^0(\mathcal{P}(X), \mathbb{R})$ such that for all $f \in \mathcal{F}$ and $\epsilon > 0$ there exist $B \in \mathbb{R}_{\geq 0}$ and sequences $f_M \in \mathcal{F}_M$, $\hat{f}_M \in H_M$, $M \in \mathbb{N}_+$, such that 1) $f_M \xrightarrow{\mathcal{P}_1} f$ 2) $\|f_M - \hat{f}_M\|_\infty \leq \epsilon$ for all $M \in \mathbb{N}_+$ 3) $\|\hat{f}_M\|_M \leq B$ for all $M \in \mathbb{N}_+$. Then for all $f \in \mathcal{F}$ and $\epsilon > 0$, there exists $\hat{f} \in H_k$ with $\|f - \hat{f}\|_\infty \leq \epsilon$.

Intuitively, the set $\mathcal{F}$ consists of all continuous functions on $\mathcal{P}(X)$ that arise as a mean field limit of functions which can be uniformly approximated by uniformly norm-bounded RKHS functions. The result then states (to use a somewhat imprecise terminology) that the RKHS $H_k$ is dense in $\mathcal{F}$. We can interpret this as an appropriate mean field variant of the universality property of kernels: a kernel on a compact metric space is called universal if its associated RKHS is dense w.r.t. the supremum norm in the space of continuous functions, and many common kernels are universal, cf. e.g. [20, Section 4.6]. In our setting, ideally universality of the finite-input kernels $k_M$ is inherited by the mean field limit kernel $k$. However, since the mean field limit can be interpreted as a form of smoothing limit, some uniformity requirements should be expected. Proposition 3.1 provides exactly such a condition.

**Remark 3.2.** In Proposition 3.1, the set $\mathcal{F}$ is a subvectorspace of $C^0(\mathcal{P}(X), \mathbb{R})$. Furthermore, if the $\mathcal{P}_1$-convergence in the definition of $\mathcal{F}$ is uniform, then $\mathcal{F}$ is closed.

Since $k_M$ and $k$ are kernels, we have the usual representer theorem for their corresponding RKHSs, cf. e.g. [27]. A natural question is then whether we have mean field convergence of the minimizers and their representation. This is clarified by the next result.

**Theorem 3.3.** Let $N \in \mathbb{N}_+$, $\mu_1, \dots, \mu_N \in \mathcal{P}(X)$ and for $n = 1, \dots, N$ let $\vec{x}_n^{[M]} \in X^M$, $M \in \mathbb{N}_+$, such that $\hat{\mu}[\vec{x}_n^{[M]}] \xrightarrow{d_{\mathrm{KR}}} \mu_n$ for $M \to \infty$. Let $L : \mathbb{R}^N \to \mathbb{R}_{\geq 0}$ be continuous and strictly convex and $\lambda > 0$. For each $M \in \mathbb{N}_+$ consider the problem

$$\min_{f \in H_M} L(f(\vec{x}_1^{[M]}), \dots, f(\vec{x}_N^{[M]})) + \lambda\|f\|_M, \tag{5}$$

as well as the problem

$$\min_{f \in H_k} L(f(\mu_1), \dots, f(\mu_N)) + \lambda\|f\|_k. \tag{6}$$

Then for each $M \in \mathbb{N}_+$ problem (5) has a unique solution $f_M^*$, which is of the form $f_M^* = \sum_{n=1}^N \alpha_n^{[M]} k_M(\cdot, \vec{x}_n^{[M]}) \in H_M$, with $\alpha_1^{[M]}, \dots, \alpha_N^{[M]} \in \mathbb{R}$, and problem (6) has a unique solution $f^*$, which is of the form $f^* = \sum_{n=1}^N \alpha_n k(\cdot, \mu_n) \in H_k$, with $\alpha_1, \dots, \alpha_N \in \mathbb{R}$. Furthermore, there exists a subsequence $(f_{M_\ell}^*)_\ell$ such that $f_{M_\ell}^* \xrightarrow{\mathcal{P}_1} f^*$ and

$$L(f_{M_\ell}^*(\vec{x}_1^{[M_\ell]}), \dots, f_{M_\ell}^*(\vec{x}_N^{[M_\ell]})) + \lambda\|f_{M_\ell}^*\|_{M_\ell} \to L(f^*(\mu_1), \dots, f^*(\mu_N)) + \lambda\|f^*\|_k. \tag{7}$$

for $\ell \to \infty$.

The main point of this result is the convergence of the minimizers, which we will establish using a $\Gamma$-convergence argument. This approach seems to have been introduced by [28, 18, 29] originally in the context of multi-agent systems.

**Remark 3.4.** An inspection of the proof reveals that in Theorem 3.3 we can replace the term $\lambda\|\cdot\|_M$ and $\lambda\|\cdot\|_k$ by $\Omega(\|\cdot\|_M)$ and $\Omega(\|\cdot\|_k)$, where $\Omega : \mathbb{R}_{\geq 0} \to \mathbb{R}_{\geq 0}$ is a nonnegative, strictly increasing and continuous function.

## 4 Support Vector Machines with mean field limit kernels

We now turn to the mean field limit of kernels in the context of statistical learning theory, focusing on SVMs. We first briefly recall the standard setup of statistical learning theory, and formulate an appropriate mean field limit thereof. We then investigate empirical and infinite-sample solutions of SVMs and their mean field limits, as well as the convergence of the corresponding risks.

**Statistical learning theory setup** We now introduce the standard setup of statistical learning theory, following mostly [20, Chapters 2 and 5]. Let $\mathcal{X} \neq \emptyset$ (associated with some $\sigma$-algebra) and $\emptyset \neq Y \subseteq \mathbb{R}$ closed (associated with the corresponding Borel $\sigma$-algebra). A *loss function* is in this setting a measurable function $\ell : \mathcal{X} \times Y \times \mathbb{R} \to \mathbb{R}_{\geq 0}$. Let $P$ be a probability distribution on $\mathcal{X} \times Y$ and $f : \mathcal{X} \to \mathbb{R}$ a measurable function, then the *risk of $f$ w.r.t. $P$ and loss function $\ell$* is defined by

$$\mathcal{R}_{\ell,P}(f) = \int_{\mathcal{X} \times Y} \ell(x, y, f(x)) \mathrm{d}P.$$

Note that this is always well-defined since $(x, y) \mapsto \ell(x, y, f(x))$ is a measurable and nonnegative function. For a set $H \subseteq \mathbb{R}^{\mathcal{X}}$ of measurable functions we also define the *minimal risk over $H$* by

$$\mathcal{R}_{\ell,P}^{H*} = \inf_{f \in H} \mathcal{R}_{\ell,P}(f).$$

If $H$ is a normed vector space, we additionally define the *regularized risk of $f \in H$* and the *minimal regularized risk over $H$* by

$$\mathcal{R}_{\ell,P,\lambda}(f) = \mathcal{R}_{\ell,P}(f) + \lambda \|f\|_H^2, \qquad \mathcal{R}_{\ell,P,\lambda}^{H*} = \inf_{f \in H} \mathcal{R}_{\ell,P,\lambda}(f),$$

where $\lambda \in \mathbb{R}_{>0}$ is the *regularization parameter*. A *data set of size $N \in \mathbb{N}_+$* is a tuple $D_N = ((x_1, y_1), \ldots, (x_N, y_N)) \in (\mathcal{X} \times Y)^N$ and for a function $f : \mathcal{X} \to \mathbb{R}$ we define its *empirical risk* by

$$\mathcal{R}_{\ell,D_N}(f) = \frac{1}{N} \sum_{n=1}^{N} \ell(x_n, y_n, f(x_n)).$$

If $H$ is a normed vector space and $f \in H$, we define additionally the *regularized empirical risk* and the *minimal regularized empirical risk over $H$* by

$$\mathcal{R}_{\ell,D_N,\lambda}(f) = \mathcal{R}_{\ell,D_N}(f) + \lambda \|f\|_H^2, \qquad \mathcal{R}_{\ell,D_N,\lambda}^{H*} = \inf_{f \in H} \mathcal{R}_{\ell,D_N,\lambda}(f),$$

where $\lambda \in \mathbb{R}_{>0}$ is again the regularization parameter. Note that the notation for the empirical risks is consistent with the risk w.r.t. a probability distribution $P$, if we identify a data set $D_N$ by the corresponding empirical distribution $\frac{1}{N} \sum_{n=1}^{N} \delta_{(x_n, y_n)}$.

In the following, $H$ will be a RKHS and a minimizer (assuming existence and uniqueness) of $\mathcal{R}_{\ell,P,\lambda}^{H*}$ will be called an *infinite-sample support vector machine (SVM)*. Similarly, $\mathcal{R}_{\ell,D_N,\lambda}^{H*}$ will be called the *empirical solution of the SVM w.r.t. the data set $D_N$*. Note that this is the common terminology in statistical learning theory, cf. [20], and corresponds to (empirical) risk minimization with Tikhonov regularization.

**Statistical learning theory setup in the mean field limit** Let now $\emptyset \neq Y \subseteq \mathbb{R}$ be compact and $\ell_M : X^M \times Y \times \mathbb{R} \to \mathbb{R}_{\geq 0}$, $M \in \mathbb{N}$, such that 1) $\ell_M(\sigma \vec{x}, y, t) = \ell_M(\vec{x}, y, t)$ for all $\vec{x} \in X^M$, $\sigma \in \mathcal{S}_M, y \in Y, t \in \mathbb{R}$; 2) there exists $C_\ell \in \mathbb{R}_{\geq 0}$ and a nondecreasing function $b : \mathbb{R}_{\geq 0} \to \mathbb{R}_{\geq 0}$ with $|\ell_M(\vec{x}, y, t)| \leq C_\ell + b(|t|)$ for all $M \in \mathbb{N}$ and $\vec{x} \in X^M, y \in Y, t \in \mathbb{R}$; 3) there exists $L_\ell \in \mathbb{R}_{\geq 0}$ with

$$|\ell_M(\vec{x}_1, y_1, t_1) - \ell_M(\vec{x}_2, y_2, t_2)| \leq L_\ell(d_{\mathrm{KR}}(\hat{\mu}[\vec{x}_1], \hat{\mu}[\vec{x}_2]) + |y_1 - y_2| + |t_1 - t_2|)$$

for all $\vec{x}_1, \vec{x}_2 \in X^M, y_1, y_1' \in Y, t_1, t_2 \in \mathbb{R}$. In particular, all $\ell_M$ are measurable (assuming the Borel $\sigma$-algebra on $X^M$) and hence are loss functions on $X^M \times Y$. Proposition 2.1 ensures the existence of a subsequence $(\ell_{M_m})_m$ and an $L_\ell$-Lipschitz continuous function $\ell : \mathcal{P}(X) \times Y \times \mathbb{R} \to \mathbb{R}$ with

$$\lim_{M \to \infty} \sup_{\substack{\vec{x} \in X^{M_m} \\ y \in Y, t \in K}} |\ell_{M_m}(\vec{x}, y, t) - \ell(\hat{\mu}[\vec{x}], y, t)| = 0 \tag{8}$$

for all compact $K \subseteq \mathbb{R}$, and we write again $\ell_{M_m} \xrightarrow{\mathcal{P}_1} \ell$. *For readability, from now on we switch to this subsequence.* Furthermore, we also get from Proposition 2.1 that there exists some $C_L \in \mathbb{R}_{\geq 0}$ such that $|\ell(\mu, y, t)| \leq C_L + b(|t|)$ for all $\mu \in \mathcal{P}(X), y \in Y, t \in \mathbb{R}$.

**Remark 4.1.** Note that, for Proposition 2.1 to apply, it is enough to assume in item 2) above the existence of a function $b : \mathbb{R} \to \mathbb{R}_{\geq 0}$ with $|\ell_M(\vec{x}, y, t)| \leq C_\ell + b(|t|)$. However, we chose the slightly stronger condition that $b$ is nondecreasing, since then $\ell_M$ is a *Nemitskii loss* according to [20, Definition 2.16]. Since the function with constant value $C_\ell$ is actually $P_M$-integrable, this means that $\ell_M$ is even a $P_M$-*integrable Nemitskii loss* according to [20]. A similar remark then applies to $\ell$.

**Lemma 4.2.** The function $\ell$ is nonnegative. Furthermore, if all $\ell_M$ are convex loss functions [20, Definition 2.12], i.e., if for all $M \in \mathbb{N}_+, \vec{x} \in X^M, y \in Y, t_1, t_2 \in \mathbb{R}$ and $\lambda \in (0,1)$ we have

$$\ell_M(\vec{x}, y, \lambda t_1 + (1-\lambda)t_2) \le \lambda \ell_M(\vec{x}, y, t_1) + (1-\lambda)\ell_M(\vec{x}, y, t_2), \tag{9}$$

then so is $\ell$.

**Empirical SVM solutions**  Given data sets $D_N^{[M]} = \left( (\vec{x}_1^{[M]}, y_1^{[M]}), \ldots, (\vec{x}_N^{[M]}, y_N^{[M]}) \right)$ for all $M \in \mathbb{N}_+$ with $\vec{x}_n^{[M]} \in X^M$, $y_n^{[M]} \in Y$, and $D_N = ((\mu_1, y_1), \ldots, (\mu_N, y_N))$ with $\mu_n \in \mathcal{P}(X)$ and $y_n \in Y$, we write $D_N^{[M]} \xrightarrow{\mathcal{P}_1} D_N$ if $\hat{\mu}[\vec{x}_n^{[M]}] \xrightarrow{d_{\mathrm{KR}}} \mu_n$ and $y_n^{[M]} \to y_n$ (where $M \to \infty$) for all $n = 1, \ldots, N$. We can interpret this as mean field convergence of the data sets.

Furthermore, consider the empirical risk of hypothesis $f_M \in H_M$ (and $f \in H_k$) on data set $D_N^{[M]}$ (and $D_N$)

$$\mathcal{R}_{\ell_M, D_N^{[M]}}(f_M) = \frac{1}{N} \sum_{n=1}^N \ell_M(\vec{x}_n^{[M]}, y_n^{[M]}, f_M(\vec{x}_n^{[M]})), \qquad \mathcal{R}_{\ell, D_N}(f) = \frac{1}{N} \sum_{n=1}^N \ell(\mu_n, y_n, f(\mu_n)),$$

and the corresponding regularized risk

$$\mathcal{R}_{\ell_M, D_N^{[M]}, \lambda}(f_M) = \frac{1}{N} \sum_{n=1}^N \ell_M(\vec{x}_n^{[M]}, y_n^{[M]}, f_M(\vec{x}_n^{[M]})) + \lambda \|f_M\|_M^2$$

$$\mathcal{R}_{\ell, D_N, \lambda}(f) = \frac{1}{N} \sum_{n=1}^N \ell(\mu_n, y_n, f(\mu_n)) + \lambda \|f\|_k^2,$$

where $\lambda \in \mathbb{R}_{>0}$ is the regularization parameter.

**Proposition 4.3.** Let $\lambda > 0$, assume that all $\ell_M$ are convex and let $D_N^{[M]}$, $D_N$ be finite data sets with $D_N^{[M]} \xrightarrow{\mathcal{P}_1} D_N$. Then for all $M \in \mathbb{N}_+$, $H_M \ni f_M \mapsto \mathcal{R}_{\ell_M, D_N^{[M]}, \lambda}(f_M)$ has a unique minimizer $f_{M,\lambda}^* \in H_M$ and $H_k \ni f \mapsto \mathcal{R}_{\ell, D_N, \lambda}(f)$ has a unique minimizer $f_\lambda^* \in H_k$. Furthermore, for all $M \in \mathbb{N}_+$ there exist $\alpha_n^{[M]} \in \mathbb{R}$, $n = 1, \ldots, N$, such that $f_{M,\lambda}^* = \sum_{n=1}^N \alpha_n^{[M]} k_M(\cdot, \vec{x}_n^{[M]})$, and there exist $\alpha_1, \ldots, \alpha_N \in \mathbb{R}$ such that $f_\lambda^* = \sum_{n=1}^N \alpha_n k(\cdot, \mu_n)$. Finally, there exists a subsequence $(f_{M_m, \lambda}^*)_m$ such that $f_{M_m, \lambda}^* \xrightarrow{\mathcal{P}_1} f_\lambda^*$ and $\mathcal{R}_{\ell_{M_m}, D_N^{[M_m]}, \lambda}(f_{M_m, \lambda}^*) \to \mathcal{R}_{\ell, D_N, \lambda}(f_\lambda^*)$ for $m \to \infty$.

**Convergence of distributions and infinite-sample SVMs in the mean field limit**  We now turn to the question of mean field limits of distributions and the associated learning problems and SVM solutions. Let $(P^{[M]})_M$ be a sequence of distributions, where $P^{[M]}$ is a probability distribution on $X^M \times Y$, and let $P$ be a probability distribution on $\mathcal{P}(X) \times Y$. We say that $P^{[M]}$ *converges in mean field to* $P$ and write $P^{[M]} \xrightarrow{\mathcal{P}_1} P$, if for all continuous (w.r.t. the product topology on $\mathcal{P}(X) \times Y$) and bounded [4] $f$ we have

$$\int_{X^M \times Y} f(\hat{\mu}[\vec{x}], y) \, \mathrm{d}P^{[M]}(\vec{x}, y) \to \int_{\mathcal{P}(X) \times Y} f(\mu, y) \, \mathrm{d}P(\mu, y). \tag{10}$$

This convergence notion of probability distributions (on different input spaces) appears to be not standard, but it is a natural concept in the present context. Essentially, it is weak (also called narrow) convergence of probability distributions adapated to our setting.

Consider now data sets $D_N^{[M]}$, $D_N$, with $D_N^{[M]} \xrightarrow{\mathcal{P}_1} D_N$, then we also have convergence in mean field of the datasets, interpreted as empirical distributions: let $f \in C^0(\mathcal{P}(X) \times Y, \mathbb{R})$ be bounded, then

$$\int_{X^M \times Y} f(\hat{\mu}[\vec{x}], y) \, \mathrm{d}D_N^{[M]}(\vec{x}, y) = \frac{1}{N} \sum_{n=1}^N f(\hat{\mu}[\vec{x}_n^{[M]}], y_n^{[M]})$$

$$\xrightarrow{M \to \infty} \frac{1}{N} \sum_{n=1}^N f(\mu_n, y_n) = \int_{\mathcal{P}(X) \times Y} f(\mu, y) \, \mathrm{d}D_N(\mu, y).$$

---

[4]Of course, since $Y$ is compact, all continuous $f$ are bounded in our present setting.

This shows that the mean field convergence of probability distributions as defined here is a direct generalization of the natural notion of mean field convergence of data sets.

Finally, consider the risk of hypothesis $f_M \in H_M$ and $f \in H_k$ w.r.t. the distribution $P^{[M]}$ and $P$, respectively,

$$\mathcal{R}_{\ell_M, P^{[M]}}(f_M) = \int_{X^M \times Y} \ell_M(\vec{x}, y, f_M(\vec{x})) \mathrm{d}P^{[M]}(\vec{x}, y)$$

$$\mathcal{R}_{\ell, P}(f) = \int_{\mathcal{P}(X) \times Y} \ell(\mu, y, f(\mu)) \mathrm{d}P(\mu, y),$$

as well as the minimal risks

$$\mathcal{R}_{\ell_M, P^{[M]}}^{H_M *} = \inf_{f_M \in H_M} \mathcal{R}_{\ell_M, P^{[M]}}(f_M) \qquad \mathcal{R}_{\ell, P}^{H_k *} = \inf_{f \in H_k} \mathcal{R}_{\ell, P}(f).$$

Our first result ensures that mean field convergence of distributions $P^{[M]}$, loss functions $\ell_M$ and data sets $D_N^{[M]}$ ensures the convergence of the corresponding risks of the empirical SVM solutions.

**Lemma 4.4.** Consider the situation and notation of Proposition 4.3 and assume that $P^{[M]} \xrightarrow{\mathcal{P}_1} P$. We then have $\mathcal{R}_{\ell_{M_m}, P^{[M_m]}}(f_{M_m, \lambda}^*) \to \mathcal{R}_{\ell, P}(f_\lambda^*)$ for $m \to \infty$.

Next, we investigate the mean field convergence of infinite-sample SVM solutions and their associated risks. Define for $\lambda \in \mathbb{R}_{\geq 0}$ (and all $M \in \mathbb{N}_+$) the regularized risk of $f_M \in H_M$ and $f \in H_k$, respectively, by

$$\mathcal{R}_{\ell_M, P^{[M]}, \lambda}(f_M) = \mathcal{R}_{\ell_M, P^{[M]}}(f_M) + \lambda \|f_M\|_M^2, \qquad \mathcal{R}_{\ell, P, \lambda}(f) = \mathcal{R}_{\ell, P}(f) + \lambda \|f\|_k^2,$$

and the corresponding minimal risks by

$$\mathcal{R}_{\ell_M, P^{[M]}, \lambda}^{H_M *} = \inf_{f_M \in H_M} \mathcal{R}_{\ell_M, P^{[M]}, \lambda}(f_M), \qquad \mathcal{R}_{\ell, P, \lambda}^{H_k *} = \inf_{f \in H_k} \mathcal{R}_{\ell, P, \lambda}(f).$$

**Proposition 4.5.**[5] Let $\lambda > 0$, assume that all $\ell_M$ are convex loss functions and let $P^{[M]}$ and $P$ be probability distributions on $X^M \times Y$ and $\mathcal{P}(X) \times Y$, respectively, with $P^{[M]} \xrightarrow{\mathcal{P}_1} P$. Then for all $M \in \mathbb{N}_+$, $H_M \ni f_M \mapsto \mathcal{R}_{\ell_M, P^{[M]}, \lambda}(f_M)$ has a unique minimizer $f_{M, \lambda}^* \in H_M$ and $H_k \ni f \mapsto \mathcal{R}_{\ell, P, \lambda}(f)$ has a unique minimizer $f_\lambda^* \in H_k$. Furthermore, there exists a subsequence $(f_{M_m, \lambda}^*)_m$ such that $f_{M_m, \lambda}^* \xrightarrow{\mathcal{P}_1} f_\lambda^*$ and $\mathcal{R}_{\ell_{M_m}, P^{[M_m]}, \lambda}(f_{M_m, \lambda}^*) \to \mathcal{R}_{\ell, P, \lambda}(f_\lambda^*)$ for $m \to \infty$. In particular, $\mathcal{R}_{\ell_{M_m}, P^{[M_m]}, \lambda}^{H_{M_m} *} \to \mathcal{R}_{\ell, P, \lambda}^{H_k *}$.

Finally, we would like to show that $\mathcal{R}_{\ell_M, P^{[M]}}^{H_M *} \to \mathcal{R}_{\ell, P}^{H_k *}$ for $P^{[M]} \xrightarrow{\mathcal{P}_1} P$. Up to a subsequence, this is established under Assumption 4.6. Define the *approximation error functions*, cf. [20, Definition 5.14], by

$$A_2^{[M]}(\lambda) = \inf_{f \in H_M} \mathcal{R}_{\ell_M, P^{[M]}, \lambda}(f) - \mathcal{R}_{\ell_M, P^{[M]}}^{H_M *} \qquad A_2(\lambda) = \inf_{f \in H_k} \mathcal{R}_{\ell, P, \lambda}(f) - \mathcal{R}_{\ell, P}^{H_k *},$$

where $M \in \mathbb{N}_+$ and $\lambda \in \mathbb{R}_{\geq 0}$. Note that (for all $M \in \mathbb{N}_+$) $A_2^{[M]}, A_2 : \mathbb{R}_{\geq 0} \to \mathbb{R}_{\geq 0}$ are increasing, concave and continuous, and $A_2^{[M]}, A_2(0) = 0$, cf. [20, Lemma 5.15]. We need essentially equicontinuity of $(A_2^{[M]})_M$ in 0, which is formalized in the following assumption.

**Assumption 4.6.** For all $\epsilon > 0$ there exists $\lambda_\epsilon > 0$ such that for all $0 < \lambda \leq \lambda_\epsilon$ and $M \in \mathbb{N}_+$ we have $A_2^{[M]}(\lambda) \leq \epsilon$.

**Proposition 4.7.** Assume that all $\ell_M$ are convex loss functions, let $P^{[M]}$ and $P$ be probability distributions on $X^M \times Y$ and $\mathcal{P}(X) \times Y$, respectively, with $P^{[M]} \xrightarrow{\mathcal{P}_1} P$. If Assumption 4.6 holds, there exists a strictly increasing sequence $(M_m)_m$ with $\mathcal{R}_{\ell_{M_m}, P^{[M_m]}}^{H_{M_m} *} \to \mathcal{R}_{\ell, P}^{H_k *}$ for $m \to \infty$.

---

[5] Note that Proposition 4.3 is actually a corollary of this result. However, since the former result is independent of the notion of mean field convergence of probability distributions, we stated and proved it separately.

# 5 Conclusion

We investigated the mean field limit of kernels and their RKHSs, as well as the mean field limit of statistical learning problems solved with SVMs. In particular, we managed to complete the basic theory of mean field kernels as started in [22]. Additionally, we investigated their approximation capabilities by providing a first approximation result and a variant of the representer theorem for mean field kernels. Finally, we introduced a corresponding mean field limit of statistical learning problems and provided convergence results for SVMs using mean field kernels. In contrast to other settings involving a large number of variables, for example, infinite-width neural networks, here we considered the case of an increasing number of inputs. This work opens many directions for future investigation. For example, it would be interesting to remove or weaken Assumption 4.6 for a result like Proposition 4.7. Another relevant direction is to find approximation results that are stronger than Proposition 3.1. Finally, it would be interesting to investigate whether statistical guarantees, like consistency or learning rates, for the finite-input learning problems can be transferred to the mean field level.

## Acknowledgments and Disclosure of Funding

We would like to thank Noel Brindise, Pierre-François Massiani and Alexander von Rohr for very detailed and helpful comments on the manuscript, and the anonymous reviewers for their detailed and helpful comments. This work is funded in part under the Excellence Strategy of the Federal Government and the Länder (G:(DE-82)EXS-SF-SFDdM035), which the authors gratefully acknowledge. The authors further thank the Deutsche Forschungsgemeinschaft (DFG, German Research Foundation) for the financial support through 320021702/GRK2326, 333849990/IRTG-2379, B04, B05, and B06 of 442047500/SFB1481, HE5386/18-1,19-2,22-1,23-1,25-1.

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

# Supplementary Material

## A Proofs

In this section of the supplementary material, we provide detailed proofs for all results in the main text.

### A.1 Proofs for Section 2

We start with Proposition 2.1, whose proof is based on [26, Lemma 1.2].

*Proof. of Proposition 2.1* For $M \in \mathbb{N}_+$ define the McShane extension $F_M : \mathcal{P}(X) \times Z \to \mathbb{R}$ by

$$F_M(\mu, z) = \inf_{\vec{x} \in X^M} f_M(\vec{x}, z) + L_f d_{\mathsf{KR}}(\hat{\mu}[\vec{x}], \mu).$$

Observe that $F_M$ is well-defined (i.e., $\mathbb{R}$-valued) since $f_M(\cdot, z)$ and $L_f d_{\mathsf{KR}}(\hat{\mu}[\cdot], \mu)$ are bounded for every $z \in Z$ (since $f_M$ and $d_{\mathsf{KR}}(\hat{\mu}[\cdot], \mu)$ are continuous and $\mathcal{P}(X)$ is compact, hence bounded).

**Step 1** $F_M$ extends $f_M$, i.e., for all $M \in \mathbb{N}_+$, $\vec{x} \in X^M$ and $z \in Z$ we have $F_M(\hat{\mu}[\vec{x}], z) = f_M(\vec{x}, z)$. To show this, let $\vec{x} \in X^M$ and $z \in Z$ be arbitrary and observe that by definition

$$F_M(\hat{\mu}[\vec{x}], z) = \inf_{\vec{x}' \in X^M} f_M(\vec{x}', z) + L_f d_{\mathsf{KR}}(\hat{\mu}[\vec{x}'], \hat{\mu}[\vec{x}]) \le f_M(\vec{x}, z) + L_f d_{\mathsf{KR}}(\hat{\mu}[\vec{x}], \hat{\mu}[\vec{x}]) = f_M(\vec{x}, z).$$

If $F_M(\hat{\mu}[\vec{x}], z) < f_M(\vec{x}, z)$, then there exists some $\vec{x}' \in X^M$ such that

$$f_M(\vec{x}', z) + L_f d_{\mathsf{KR}}(\hat{\mu}[\vec{x}'], \hat{\mu}[\vec{x}]) < f_M(\vec{x}, z),$$

but this means that

$$L_f d_{\mathsf{KR}}(\hat{\mu}[\vec{x}'], \hat{\mu}[\vec{x}]) < f_M(\vec{x}, z) - f_M(\vec{x}', z) \le |f_M(\vec{x}, z) - f_M(\vec{x}', z)|,$$

contradicting the $L_f$-Lipschitz continuity of $f_M$.

**Step 2** All $F_M$ are $L_f$-continuous: Let $M \in \mathbb{N}_+$, $\mu_i \in \mathcal{P}(X)$ and $z_i \in Z$, $i = 1, 2$, be arbitrary. Since $X^M$ is compact and $f_M(\cdot, z)$ and $L_f d_{\mathsf{KR}}(\hat{\mu}[\cdot], \mu_i)$, $i = 1, 2$, are continuous, the infimum in the definition of $F_M$ is actually attained. Let $\vec{x}_2 \in X^M$ such that $F_M(\mu_2, z_2) = f_M(\vec{x}_2, z_2) + L_f d_{\mathsf{KR}}(\hat{\mu}[\vec{x}_2], \mu_2)$, then we have

$$
\begin{aligned}
F_M(\mu_1, z_1) &\le f_M(\vec{x}_2, z_1) + L_f d_{\mathsf{KR}}(\hat{\mu}[\vec{x}_2], \mu_1) \\
&= f_M(\vec{x}_2, z_1) + L_f d_{\mathsf{KR}}(\hat{\mu}[\vec{x}_2], \mu_2) - L_f d_{\mathsf{KR}}(\hat{\mu}[\vec{x}_2], \mu_2) + L_f d_{\mathsf{KR}}(\hat{\mu}[\vec{x}_2], \mu_1) \\
&\le f_M(\vec{x}_2, z_2) + L_f d_{\mathsf{KR}}(\hat{\mu}[\vec{x}_2], \mu_2) + L_f d_Z(z_1, z_2) - L_f d_{\mathsf{KR}}(\hat{\mu}[\vec{x}_2], \mu_2) \\
&\quad + L_f d_{\mathsf{KR}}(\hat{\mu}[\vec{x}_2], \mu_1) \\
&\le F_M(\mu_2, z_2) + L_f d_Z(z_1, z_2) - L_f d_{\mathsf{KR}}(\hat{\mu}[\vec{x}_2], \mu_2) + L_f d_{\mathsf{KR}}(\mu_1, \mu_2) \\
&\quad + L_f d_{\mathsf{KR}}(\hat{\mu}[\vec{x}_2], \mu_2) \\
&= F_M(\mu_2, z_2) + L_f(d_{\mathsf{KR}}(\mu_1, \mu_2) + d_Z(z_1, z_2)),
\end{aligned}
$$

where we used the definition of $F_M$ in the first inequality, the Lipschitz continuity of $f_M$ (w.r.t. the second argument) for the second inequality, and then the fact that $\vec{x}_2$ attains the infimum in the definition of $F_M(\mu_2, z_2)$ and the triangle inequality for $d_{\mathsf{KR}}$. Interchanging the roles of $\mu_1, z_1$ and $\mu_2, z_2$ then establishes the claim.

**Step 3** There exists $B_F \in \mathbb{R}_{\ge 0}$ such that for all $M \in \mathbb{N}_+$, $\mu \in \mathcal{P}(X)$ and $z \in Z$ we have $|F_M(\mu, z)| \le B_F + h(z)$: Let $D_{\mathcal{P}(X)}$ be the diameter of $\mathcal{P}(X)$ (which is finite since $\mathcal{P}(X)$ is compact), then for all $M \in \mathbb{N}_+$ and $\vec{x} \in X^M$, $z \in Z$, $\mu \in \mathcal{P}(X)$ we have

$$-(B_f + L_f D_{\mathcal{P}(X)} + b(z)) \le f_M(\vec{x}, z) + L_f d_{\mathsf{KR}}(\hat{\mu}[\vec{x}], \mu) \le B_f + L_f D_{\mathcal{P}(X)} + b(z),$$

therefore $|F_M(\mu, z)| \le B_f + L_f D_{\mathcal{P}(X)} + b(z)$, showing the claim with $B_F = B_f + L_f D_{\mathcal{P}(X)}$.

**Step 4** Summarizing, $(F_M)_M$ is a sequence of $L_f$-Lipschitz continuous and hence equicontinuous functions such that for all $\mu \in \mathcal{P}(X)$ and $z \in Z$, the set $\{F_M(\mu, z) \mid M \in \mathbb{N}_+\}$ is relatively compact (since it is a bounded subset of $\mathbb{R}$). We can now use a variant of the Arzela-Ascoli theorem, cf. [30,

Corollary III.3.3]. From the assumption on $Z$, we can find a sequence $(V_n)_n$ of open subsets of $Z$ such that all $\bar{V}_n$ are compact, $\bar{V}_n \subseteq V_{n+1}$ and we have $\bigcup_n V_n = Z$. Then $(F_M|_{\bar{V}_n})_M$ is a sequence of functions that fulfills the conditions of the Arzela-Ascoli theorem (since $\mathcal{P}(X) \times K_n$ is compact), so there exists a subsequence $(F_{M_\ell^{(n)}}|_{\bar{V}_n})_\ell$ that converges uniformly to a continuous function on $\mathcal{P}(X) \times \bar{V}_n$. Denote the diagonal subsequence of all these subsequences by $(F_{M_\ell})_\ell$, then there exists a continuous $f : \mathcal{P}(X) \times Z \to \mathbb{R}$ such that $(F_{M_\ell})_\ell$ converges uniformly on compact subsets to $f$. Since $\mathcal{P}(X)$ is compact, this means that for all compact $K \subseteq Z$

$$\lim_\ell \sup_{\substack{\mu \in \mathcal{P}(X) \\ z \in K}} |F_{M_\ell}(\mu, z) - f(\mu, z)| = 0.$$

This also implies that for all $\mu \in \mathcal{P}(X)$ and $z \in Z$ we have $|f(\mu, z)| \leq B_F + b(z)$.

Furthermore, $f$ is also $L_f$-Lipschitz continuous: Let $\mu_i \in \mathcal{P}(X)$, $z_i \in Z$, $i = 1, 2$, and $\epsilon > 0$ be arbitrary. Let $K \subseteq Z$ be compact with $z_1, z_2 \in K$ and choose $\ell \in \mathbb{N}_+$ such that

$$\sup_{\substack{\mu \in \mathcal{P}(X) \\ z \in K}} |F_{M_\ell}(\mu, z) - f(\mu, z)| \leq \frac{\epsilon}{2}.$$

We then have

$$\begin{aligned} |f(\mu_1, z_1) - f(\mu_2, z_2)| &\leq |f(\mu_1, z_1) - F_{M_\ell}(\mu_1, z_1)| + |F_{M_\ell}(\mu_1, z_1) - F_{M_\ell}(\mu_2, z_2)| \\ &\quad + |F_{M_\ell}(\mu_2, z_2) - f(\mu_2, z_2)| \\ &\leq L_f \left( d_{\mathrm{KR}}(\mu_1, \mu_2) + d_Z(z_1, z_2) \right) + \epsilon, \end{aligned}$$

and since $\epsilon > 0$ was arbitrary, the claim follows.

**Step 5** For $\ell \in \mathbb{N}_+$ and $\vec{x} \in X^{M_\ell}$, $z \in Z$ we have

$$|f_{M_\ell}(\vec{x}, z) - f(\hat{\mu}[\vec{x}], z)| = |F_{M_\ell}(\hat{\mu}[\vec{x}], z) - f(\hat{\mu}[\vec{x}], z)|$$

since $F_{M_\ell}$ extends $f_{M_\ell}$, and hence

$$\sup_{\substack{\vec{x} \in X^{M_\ell} \\ z \in K}} |f_{M_\ell}(\vec{x}, z) - f(\hat{\mu}[\vec{x}], z)| \to 0.$$

$\square$

Next, we provide the proofs for the $\Gamma\text{-}\liminf$ and $\Gamma\text{-}\limsup$ results.

*Proof. of Lemma 2.4* Assume the statement is not true, i.e., $\|f\|_k > \liminf_{M \to \infty} \|f_M\|_M$. This means that there exists a subsequence $M_\ell$ and $C \in \mathbb{R}_{\geq 0}$ such that $\|f\|_k > \lim_\ell \|f_{M_\ell}\|_{M_\ell} = C$. Note that this implies that $\|f\|_k > 0$.

Let $\epsilon_1, \epsilon_2 > 0$ and $\alpha > 1$, $\beta \in (0, 1)$ be arbitrary. From Theorem B.1, there exists $(\vec{\mu}, \vec{\alpha}) \in \mathcal{P}(X)^N \times \mathbb{R}^N$ such that

$$\mathcal{D}(\vec{\mu}, \vec{\alpha}, f, k) + \epsilon_1 \geq \|f\|_k,$$

and w.l.o.g. we can assume that $\epsilon_1 > 0$ is small enough so that $\mathcal{D}(\vec{\mu}, \vec{\alpha}, f, k) > 0$. The latter implies that $\mathcal{E}(\vec{\mu}, \vec{\alpha}, f)$, $\mathcal{W}(\vec{\mu}, \vec{\alpha}, k) > 0$, so defining

$$\epsilon_\alpha = \frac{\alpha - 1}{\alpha} \mathcal{E}(\vec{\mu}, \vec{\alpha}, f)$$

$$\epsilon_\beta = (1/\beta - 1) \mathcal{W}(\vec{\mu}, \vec{\alpha}, k)$$

we get $\epsilon_\alpha, \epsilon_\beta > 0$. For each $n = 1, \dots, N$, choose $\vec{x}_n^{[M]} \in X^M$ such that $\vec{x}_n^{[M]} \xrightarrow{d_{\mathrm{KR}}} \mu_n$ for $M \to \infty$. Choose now $L_1 \in \mathbb{N}$ such that for all $\ell \geq L_1$ we get

$$|\mathcal{E}(\vec{X}^{[M_\ell]}, \vec{\alpha}, f_{M_\ell}) - \mathcal{E}(\vec{\mu}, \vec{\alpha}, f)| \leq \epsilon_\alpha$$

$$|\mathcal{W}(\vec{X}^{[M_\ell]}, \vec{\alpha}, k_{M_\ell}) - \mathcal{W}(\vec{\mu}, \vec{\alpha}, k)| \leq \epsilon_\beta.$$

(cf. also the proof of Theorem 2.3) and $\mathcal{W}(\vec{X}^{[M_\ell]}, \vec{\alpha}, k^{[M_\ell]}) > 0$. We then get

$$\mathcal{E}(\vec{\mu}, \vec{\alpha}, f) \leq \alpha \mathcal{E}(\vec{X}^{[M_\ell]}, \vec{\alpha}, f_{M_\ell})$$

$$\mathcal{W}(\vec{\mu}, \vec{\alpha}, k) \geq \beta \mathcal{W}(\vec{X}^{[M_\ell]}, \vec{\alpha}, k^{[M_\ell]}),$$

so altogether

$$\frac{\mathcal{E}(\vec{\mu}, \vec{\alpha}, f)}{\mathcal{W}(\vec{\mu}, \vec{\alpha}, k)} \le \frac{\alpha \mathcal{E}(\vec{X}^{[M_\ell]}, \vec{\alpha}, f_{M_\ell})}{\beta \mathcal{W}(\vec{X}^{[M_\ell]}, \vec{\alpha}, k^{[M_\ell]})}.$$

Using Theorem B.1 again leads to

$$\frac{\alpha \mathcal{E}(\vec{X}^{[M_\ell]}, \vec{\alpha}, f_M)}{\beta \mathcal{W}(\vec{X}^{[M_\ell]}, \vec{\alpha}, k^{[M_\ell]})} = \mathcal{D}(\vec{X}^{[M_\ell]}, \vec{\alpha}, f_{M_\ell}, k^{[M_\ell]}) \le \|f_{M_\ell}\|_{M_\ell}.$$

Finally, let $L_2$ such that for all $\ell \ge L_2$ we have $\|f_{M_\ell}\|_{M_\ell} \le C + \epsilon_2$. For $\ell \ge L_1, L_2$ we then get

$$\begin{aligned}
C < \|f\|_k &\le \mathcal{D}(\vec{\mu}, \vec{\alpha}, f, k) + \epsilon_1 \\
&= \frac{\mathcal{E}(\vec{\mu}, \vec{\alpha}, f)}{\mathcal{W}(\vec{\mu}, \vec{\alpha}, k)} + \epsilon_1 \\
&\le \frac{\alpha \mathcal{E}(\vec{X}^{[M_\ell]}, \vec{\alpha}, f_{M_\ell})}{\beta \mathcal{W}(\vec{X}^{[M_\ell]}, \vec{\alpha}, k^{[M_\ell]})} + \epsilon_1 \\
&\le \frac{\alpha}{\beta} \|f_{M_\ell}\|_{M_\ell} + \epsilon_1 \\
&\le \frac{\alpha}{\beta} C + \frac{\alpha}{\beta} \epsilon_2 + \epsilon_1.
\end{aligned}$$

Since $\epsilon_1, \epsilon_2 > 0$ and $\alpha > 1, \beta \in (0, 1)$ were arbitrary, this implies that

$$C < \|f\|_k \le C,$$

a contradiction. $\qquad\square$

*Proof. of Lemma 2.5* Let $f \in H_k$ be arbitrary and choose $(\epsilon_n)_n \subseteq \mathbb{R}_{>0}$ with $\epsilon_n \searrow 0$.

**Step 1** For each $n \in \mathbb{N}$ choose

$$f_n^{\mathrm{pre}} = \sum_{\ell=1}^{L_n} \alpha_\ell^{(n)} k(\cdot, \mu_\ell^{(n)}) \in H_k^{\mathrm{pre}},$$

where $\alpha_1^{(n)}, \ldots, \alpha_{L_n}^{(n)} \in \mathbb{R}$ and $\mu_1^{(n)}, \ldots, \mu_{L_n}^{(n)} \in \mathcal{P}(X)$, with

$$\|f - f_n^{\mathrm{pre}}\|_k \le \frac{\epsilon_n}{3\sqrt{C_k}}$$

and $\|f_n^{\mathrm{pre}}\|_k \le \|f\|_k$. To see that such a sequence of functions exists, choose some sequence $(\bar{f}_n)_n \in H_k^{\mathrm{pre}}$ with $\bar{f}_n = \sum_{\ell=1}^{\bar{L}_n} \bar{\alpha}_\ell^{(n)} k(\cdot, \bar{\mu}_\ell^{(n)})$, where $\bar{\alpha}_\ell^{(n)} \in \mathbb{R}$, $\bar{\mu}_\ell^{(n)} \in \mathcal{P}(X)$, with $\bar{f}_n \xrightarrow{\|\cdot\|_k} f$ (exists since $H_k^{\mathrm{pre}}$ is dense in $H_k$). Define now for $n \in \mathbb{N}$

$$\bar{H}_n = \mathrm{span}\{k(\cdot, \bar{\mu}_\ell^{(m)}) \mid m = 1, \ldots, n, \ \ell = 1, \ldots, \bar{L}_m\}$$

and $\hat{f}_n = P_{\bar{H}_n} f$, where $P_{\bar{H}_n}$ is the orthogonal projection onto $\bar{H}_n$. Then $\bar{H}_n \subseteq H_k^{\mathrm{pre}}$, $\|\hat{f}_n\|_k = \|P_{\bar{H}_n} f\|_k \le \|f\|_k$ and $\|f - \hat{f}_n\|_k \le \|f - \bar{f}_n\|_k \to 0$ (since $\hat{f}_n = P_{\bar{H}_n} f$ is the orthogonal projection of $f$ onto $\bar{H}_n$ and $\bar{f}_n \in \bar{H}_n$), hence $\hat{f}_n \xrightarrow{\|\cdot\|_k} f$. We can now choose $(f_n^{\mathrm{pre}})_n$ as a subsequence of $(\hat{f}_n)_n$.

Next, for all $n \in \mathbb{N}$ and $\ell = 1, \ldots, L_n$ choose $\vec{x}_M^{(n,\ell)} \in X^M$ with $\hat{\mu}[\vec{x}_M^{(n,\ell)}] \xrightarrow{d_{\mathrm{KR}}} \mu_\ell^{(n)}$ for $M \to \infty$. Furthermore, for all $n \in \mathbb{N}$ choose $M_n \in \mathbb{N}$ such that for all $M \ge M_n$ and $\ell = 1, \ldots, L_n$ we have

$$d_{\mathrm{KR}}(\hat{\mu}[\vec{x}_M^{(n,\ell)}], \mu_\ell^{(n)}) \le \min \left\{ \frac{\epsilon_n}{3 \left(1 + L_k \sum_{\ell'=1}^{L_n} |\alpha_{\ell'}^{(n)}|\right)}, \frac{\epsilon_n^2}{2 \left(1 + 2L_k \sum_{i,j=1}^{L_n} |\alpha_i^{(n)}||\alpha_j^{(n)}|\right)} \right\}$$

and

$$\sup_{\vec{x}, \vec{x}' \in X^M} |k_M(\vec{x}, \vec{x}') - k(\hat{\mu}[\vec{x}], \hat{\mu}[\vec{x}'])| \le \min \left\{ \frac{\epsilon_n}{3 \left(1 + \sum_{\ell'=1}^{L_n} |\alpha_{\ell'}^{(n)}|\right)}, \frac{\epsilon_n^2}{2 \left(1 + \sum_{i,j=1}^{L_n} |\alpha_i^{(n)}||\alpha_j^{(n)}|\right)} \right\}.$$

W.l.o.g. we can assume that $(M_n)_n$ is strictly increasing. For $M \in \mathbb{N}$, let $n(M)$ be the largest integer such that $M_{n(M)} \le M$ and define

$$\hat{f}_M^{\text{pre}} = \sum_{\ell=1}^{L_{n(M)}} \alpha_\ell^{(n(M))} k(\cdot, \hat{\mu}[\vec{x}_M^{(n(M)),\ell}]) \in H_k^{\text{pre}}$$

$$f_M = \sum_{\ell=1}^{L_{n(M)}} \alpha_\ell^{(n(M))} k_M(\cdot, \vec{x}_M^{(n(M)),\ell}) \in H_M^{\text{pre}}.$$

**Step 2** We now show that $f_M \xrightarrow{\mathcal{P}_1} f$. For this, let $\epsilon > 0$ be arbitrary and $n_\epsilon \in \mathbb{N}$ such that $\epsilon_n \le \epsilon$. Let now $M \ge M_{n_\epsilon}$ (note that this implies that $n(M) \ge n_\epsilon$ and hence $\epsilon_{n(M)} \le \epsilon_n$) and $\vec{x} \in X^M$, then we have

$$|f(\hat{\mu}[\vec{x}]) - f_M(\vec{x})| \le \underbrace{|f(\hat{\mu}[\vec{x}]) - f_{n(M)}(\hat{\mu}[\vec{x}])|}_{=I} + \underbrace{|f_{n(M)}(\hat{\mu}[\vec{x}]) - \hat{f}_M^{\text{pre}}(\hat{\mu}[\vec{x}])|}_{=II} + \underbrace{|\hat{f}_M^{\text{pre}}(\hat{\mu}[\vec{x}]) - f_M(\vec{x})|}_{=III}$$

We continue with

$$\begin{aligned}
I &= |f(\hat{\mu}[\vec{x}]) - f_{n(M)}(\hat{\mu}[\vec{x}])| \\
&= |\langle f - f_{n(M)}, k(\cdot, \hat{\mu}[\vec{x}]) \rangle_k| \\
&\le \|f - f_{n(M)}\|_k \|k(\cdot, \hat{\mu}[\vec{x}])\|_k \\
&= \|f - f_{n(M)}\|_k \sqrt{k(\hat{\mu}[\vec{x}], \hat{\mu}[\vec{x}])} \\
&\le \frac{\epsilon_{n(M)}}{3\sqrt{C_k}} \sqrt{C_k}
\end{aligned}$$

where we first used the reproducing property of $k$, then Cauchy-Schwarz, again the reproducing property of $k$, and finally the choice $f_{n(M)}$ and the boundedness of $k$.

Next,

$$\begin{aligned}
II &= |f_{n(M)}(\hat{\mu}[\vec{x}]) - \hat{f}_M^{\text{pre}}(\hat{\mu}[\vec{x}])| \\
&= \left| \sum_{\ell=1}^{L_{n(M)}} \alpha_\ell^{(n(M))} k(\cdot, \mu_\ell^{(n(M))}) - \sum_{\ell=1}^{L_{n(M)}} \alpha_\ell^{(n(M))} k(\cdot, \hat{\mu}[\vec{x}_M^{(n(M)),\ell}]) \right| \\
&\le \sum_{\ell=1}^{L_{n(M)}} \left| \alpha_\ell^{(n(M))} \right| |k(\cdot, \mu_\ell^{(n(M))}) - k(\cdot, \hat{\mu}[\vec{x}_M^{(n(M)),\ell}])| \\
&\le L_k \sum_{\ell=1}^{L_{n(M)}} \left| \alpha_\ell^{(n(M))} \right| d_{\text{KR}}(\hat{\mu}[\vec{x}_M^{(n(M)),\ell}], \mu_\ell^{(n(M))}) \\
&\le \frac{\epsilon_{n(M)}}{3},
\end{aligned}$$

where we used the triangle inequality, the Lipschitz continuity of $k$, and then the choice of the sequence $(M_n)_n$.

Finally,

$$\begin{aligned}
III &= |\hat{f}_M^{\text{pre}}(\hat{\mu}[\vec{x}]) - f_M(\vec{x})| \\
&= \left| \sum_{\ell=1}^{L_{n(M)}} \alpha_\ell^{(n(M))} k(\cdot, \hat{\mu}[\vec{x}_M^{(n(M)),\ell}]) - \sum_{\ell=1}^{L_{n(M)}} \alpha_\ell^{(n(M))} k_M(\cdot, \vec{x}_M^{(n(M)),\ell}) \right| \\
&\le \sum_{\ell=1}^{L_{n(M)}} \left| \alpha_\ell^{(n(M))} \right| |k(\cdot, \hat{\mu}[\vec{x}_M^{(n(M)),\ell}]) - k_M(\cdot, \vec{x}_M^{(n(M)),\ell})| \\
&\le \frac{\epsilon_{n(M)}}{3},
\end{aligned}$$

where the triangle inequality has been used in the first step and then again the choice of the sequence $(M_n)_n$.

Altogether,

$$|f(\hat{\mu}[\vec{x}]) - f_M(\vec{x})| \leq I + II + III$$
$$\leq \frac{\epsilon_{n(M)}}{3} + \frac{\epsilon_{n(M)}}{3} + \frac{\epsilon_{n(M)}}{3}$$
$$\leq \epsilon,$$

establishing $f_M \xrightarrow{\mathcal{P}_1} f$.

**Step 3** We now show $\limsup_{M\to\infty} \|f_M\|_M \leq \|f\|_k$. Let $\epsilon > 0$ be arbitrary and $n_\epsilon \in \mathbb{N}$ such that $\epsilon_n \leq \epsilon$ and let $M \geq M_{n_\epsilon}$. We have

$$\|f_M\|_M^2 = \sum_{\ell,\ell'=1}^{L_{n(M)}} \alpha_\ell^{(n(M))} \alpha_{\ell'}^{(n(M))} k_M(\vec{x}_M^{(n(M),\ell')}, \vec{x}_M^{(n(M),\ell')})$$
$$\leq \sum_{\ell,\ell'=1}^{L_{n(M)}} \alpha_\ell^{(n(M))} \alpha_{\ell'}^{(n(M))} k(\mu_{\ell'}^{(n(M))}, \mu_\ell^{(n(M))}) + |R_1| + |R_2|$$
$$= \|f_{n(M)}^{\text{pre}}\|_k^2 + R_1 + R_2$$
$$\leq \|f\|_k^2 + R_1 + R_2.$$

with remainder terms

$$R_1 = \sum_{\ell,\ell'=1}^{L_{n(M)}} \alpha_\ell^{(n(M))} \alpha_{\ell'}^{(n(M))} k_M(\vec{x}_M^{(n(M),\ell')}, \vec{x}_M^{(n(M),\ell')}) - \sum_{\ell,\ell'=1}^{L_{n(M)}} \alpha_\ell^{(n(M))} \alpha_{\ell'}^{(n(M))} k(\hat{\mu}[\vec{x}_M^{(n(M),\ell')}], \hat{\mu}[\vec{x}_M^{(n(M),\ell')}])$$

$$R_2 = \sum_{\ell,\ell'=1}^{L_{n(M)}} \alpha_\ell^{(n(M))} \alpha_{\ell'}^{(n(M))} k(\hat{\mu}[\vec{x}_M^{(n(M),\ell')}], \hat{\mu}[\vec{x}_M^{(n(M),\ell')}]) - \sum_{\ell,\ell'=1}^{L_{n(M)}} \alpha_\ell^{(n(M))} \alpha_{\ell'}^{(n(M))} k(\mu_{\ell'}^{(n(M))}, \mu_\ell^{(n(M))})$$

We now bound these terms, so that

$$R_1 = \left| \sum_{\ell,\ell'=1}^{L_{n(M)}} \alpha_\ell^{(n(M))} \alpha_{\ell'}^{(n(M))} k_M(\vec{x}_M^{(n(M),\ell')}, \vec{x}_M^{(n(M),\ell')}) - \sum_{\ell,\ell'=1}^{L_{n(M)}} \alpha_\ell^{(n(M))} \alpha_{\ell'}^{(n(M))} k(\hat{\mu}[\vec{x}_M^{(n(M),\ell')}], \hat{\mu}[\vec{x}_M^{(n(M),\ell')}]) \right|$$
$$\leq \sum_{\ell,\ell'=1}^{L_{n(M)}} |\alpha_\ell^{(n(M))}||\alpha_{\ell'}^{(n(M))}||k_M(\vec{x}_M^{(n(M),\ell')}, \vec{x}_M^{(n(M),\ell')}) - k(\hat{\mu}[\vec{x}_M^{(n(M),\ell')}], \hat{\mu}[\vec{x}_M^{(n(M),\ell')}])|$$
$$\leq \frac{\epsilon_{n(M)}^2}{2},$$

and

$$R_2 = \left| \sum_{\ell,\ell'=1}^{L_{n(M)}} \alpha_\ell^{(n(M))} \alpha_{\ell'}^{(n(M))} k(\hat{\mu}[\vec{x}_M^{(n(M),\ell')}], \hat{\mu}[\vec{x}_M^{(n(M),\ell')}]) - \sum_{\ell,\ell'=1}^{L_{n(M)}} \alpha_\ell^{(n(M))} \alpha_{\ell'}^{(n(M))} k(\mu_{\ell'}^{(n(M))}, \mu_\ell^{(n(M))}) \right|$$
$$\leq \sum_{\ell,\ell'=1}^{L_{n(M)}} |\alpha_\ell^{(n(M))}||\alpha_{\ell'}^{(n(M))}||k(\hat{\mu}[\vec{x}_M^{(n(M),\ell')}], \hat{\mu}[\vec{x}_M^{(n(M),\ell')}]) - k(\mu_{\ell'}^{(n(M))}, \mu_\ell^{(n(M))})|$$
$$\leq L_k \sum_{\ell,\ell'=1}^{L_{n(M)}} |\alpha_\ell^{(n(M))}||\alpha_{\ell'}^{(n(M))}| \left( d_{\text{KR}}(\hat{\mu}[\vec{x}_M^{(n(M),\ell)}], \mu_\ell^{(n(M))}) + d_{\text{KR}}(\hat{\mu}[\vec{x}_M^{(n(M),\ell')}], \mu_{\ell'}^{(n(M))}) \right)$$
$$\leq \frac{\epsilon_{n(M)}^2}{2}.$$

Altogether,

$$\|f_M\|_M^2 \leq \|f\|_k^2 + |R_1| + |R_2|$$
$$\leq \|f\|_k^2 + \frac{\epsilon_{n(M)}^2}{2} + \frac{\epsilon_{n(M)}^2}{2}$$
$$\leq \|f\|_k^2 + \epsilon^2,$$

so $\|f_M\|_M \leq \|f\|_k + \epsilon$ for all $M \geq M_{n_\epsilon}$, and since $\epsilon > 0$ was arbitrary, we finally get $\limsup_{M \to \infty} \|f_M\|_M \leq \|f\|_k$. $\qquad\square$

Finally, we can now provide the proof for the central Theorem 2.3.

*Proof. of Theorem 2.3* The first statement is part of Lemma 2.5. Let us turn to the second statement: The existence of the subsequence $(f_{M_\ell})_\ell$ and the continuous function $f : \mathcal{P}(X) \to \mathbb{R}$ with $f_{M_\ell} \xrightarrow{\mathcal{P}_1} f$ was shown in [22, Corollary 4.3], so we only have to ensure that $f \in H_k$ with $\|f\|_k \leq B$. For this, we use the characterization of RKHS functions from Theorem B.1. In particular, we will utilize the notation introduced there.

**Step 1** Let $(\vec{\mu}, \vec{\alpha}) \in \mathcal{P}(X)^N \times \mathbb{R}^N$. We show that if $\mathcal{W}(\vec{\mu}, \vec{\alpha}, k) = 0$, then $\mathcal{E}(\vec{\mu}, \vec{\alpha}, f) = 0$.

Assume that $\mathcal{W}(\vec{\mu}, \vec{\alpha}, k) = 0$. If $B = 0$, then $f_M \equiv 0$ and $f_{M_\ell} \xrightarrow{\mathcal{P}_1} f$ implies that $f \equiv 0$, so the claim is clear in this case. Assume now $B > 0$, let $\epsilon > 0$ be arbitary and for $n = 1, \ldots, N$, choose sequences $\vec{x}_n^{[M]} \in X^M$ such that $\vec{x}_n^{[M]} \xrightarrow{d_{\mathrm{KR}}} \mu_n$ for $M \to \infty$. For convenience, define $\vec{X}^{[M]} = \left( \vec{x}_1^{[M]} \quad \cdots \quad \vec{x}_N^{[M]} \right)$. Choose now $\ell_\epsilon \in \mathbb{N}$ such that for all $M \geq M_{\ell_\epsilon}$ we get $\mathcal{W}(\vec{X}^{[M]}, \vec{\alpha}, k_M) \leq \epsilon/B$. This is possible since $k_M \xrightarrow{\mathcal{P}_1} k$ together with the continuity of $k_M$ and $k$ as well as $\vec{x}_n^{[M]} \xrightarrow{d_{\mathrm{KR}}} \mu_n$ for $M \to \infty$ and all $n = 1, \ldots, N$ implies that $\mathcal{W}(\vec{X}^{[M]}, \vec{\alpha}, k_M) \to \mathcal{W}(\vec{\mu}, \vec{\alpha}, k) = 0$. Let now $\ell \geq \ell_\epsilon$ be arbitrary and observe that $f_M \in H_M$ implies $\mathcal{N}(f_M, k_M) < \infty$ according to Theorem B.1, so in particular $\mathcal{D}(\vec{X}^{[M_\ell]}, \vec{\alpha}, f_{M_\ell}, k_{M_\ell}) < \infty$.

If $\mathcal{W}(\vec{X}^{[M_\ell]}, \vec{\alpha}, k_{M_\ell}) = 0$, then we get that $\mathcal{E}(\vec{X}^{[M_\ell]}, \vec{\alpha}, f_{M_\ell}) = 0 \leq \epsilon$ since $\mathcal{D}(\vec{X}^{[M_\ell]}, \vec{\alpha}, f_{M_\ell}, k_{M_\ell}) < \infty$, which implies by definition that $\mathcal{E}(\vec{X}^{[M_\ell]}, \vec{\alpha}, f_{M_\ell}) = 0$.

If $\mathcal{W}(\vec{X}^{[M_\ell]}, \vec{\alpha}, k_{M_\ell}) > 0$, then we have

$$\frac{\mathcal{E}(\vec{X}^{[M_\ell]}, \vec{\alpha}, f_{M_\ell})}{\mathcal{W}(\vec{X}^{[M_\ell]}, \vec{\alpha}, k_{M_\ell})} = \mathcal{D}(\vec{X}^{[M_\ell]}, \vec{\alpha}, f_{M_\ell}, k_{M_\ell}) \leq \mathcal{N}(f_{M_\ell}, k_{M_\ell}) = \|f_{M_\ell}\|_{M_\ell} \leq B,$$

which implies

$$\mathcal{E}(\vec{X}^{[M_\ell]}, \vec{\alpha}, f_{M_\ell}) \leq B \mathcal{W}(\vec{X}^{[M_\ell]}, \vec{\alpha}, k_{M_\ell}) \leq \epsilon.$$

Since $f_{M_\ell} \xrightarrow{\mathcal{P}_1} f$ together with the continuity of $f_M$ and $f$ as well as $\vec{x}_n^{[M]} \xrightarrow{d_{\mathrm{KR}}} \mu_n$ implies that $\mathcal{E}(\vec{X}^{[M_\ell]}, \vec{\alpha}, f_{M_\ell}) \to \mathcal{E}(\vec{\mu}, \vec{\alpha}, f)$, we get that $\mathcal{E}(\vec{\mu}, \vec{\alpha}, f) \leq \epsilon$, and since $\epsilon > 0$ was arbitrary we arrive at $\mathcal{E}(\vec{\mu}, \vec{\alpha}, f) \leq 0$.

Assume now that $\mathcal{E}(\vec{\mu}, \vec{\alpha}, f) < 0$. This implies that there exist $\delta > 0$ and $\ell_\delta \in \mathbb{N}$ such that for all $\ell \geq \ell_\delta$ we have $\mathcal{E}(\vec{X}^{[M_\ell]}, \vec{\alpha}, f_{M_\ell}) \leq -\delta < 0$, since $\mathcal{E}(\vec{X}^{[M_\ell]}, \vec{\alpha}, f_{M_\ell}) \to \mathcal{E}(\vec{\mu}, \vec{\alpha}, f)$. Let $\ell \geq \ell_\delta$, then we get that $\mathcal{E}(\vec{X}^{[M_\ell]}, -\vec{\alpha}, f_{M_\ell}) \geq \delta > 0$ and we have $\mathcal{W}(\vec{X}^{[M_\ell]}, -\vec{\alpha}, k_{M_\ell}) = \mathcal{W}(\vec{X}^{[M_\ell]}, \vec{\alpha}, k_{M_\ell}) > 0$. We can then continue with

$$\frac{\delta}{\mathcal{W}(\vec{X}^{[M_\ell]}, \vec{\alpha}, k_{M_\ell})} \leq \frac{\mathcal{E}(\vec{X}^{[M_\ell]}, -\vec{\alpha}, f_{M_\ell})}{\mathcal{W}(\vec{X}^{[M_\ell]}, -\vec{\alpha}, k_{M_\ell})}$$
$$\leq \mathcal{D}(\vec{X}^{[M_\ell]}, -\vec{\alpha}, f_{M_\ell}, k_{M_\ell})$$
$$\leq \mathcal{N}(f_{M_\ell}, k_{M_\ell})$$
$$= \|f_{M_\ell}\|_{M_\ell} \leq B,$$

which implies that $\mathcal{W}(\vec{X}^{[M_\ell]}, -\vec{\alpha}, k_{M_\ell}) = \mathcal{W}(\vec{X}^{[M_\ell]}, \vec{\alpha}, k_{M_\ell}) \geq \delta/B$. But since $\mathcal{W}(\vec{X}^{[M_\ell]}, \vec{\alpha}, k_{M_\ell}) \to \mathcal{W}(\vec{\mu}, \vec{\alpha}, k)$, this implies that $\mathcal{W}(\vec{\mu}, \vec{\alpha}, k) \geq \delta/B > 0$, a contradiction. Altogether, $\mathcal{E}(\vec{\mu}, \vec{\alpha}, f) = 0$.

**Step 2** Let $(\vec{\mu}, \vec{\alpha}) \in \mathcal{P}(X)^N \times \mathbb{R}^N$. If $\mathcal{W}(\vec{\mu}, \vec{\alpha}, k) > 0$ and $\mathcal{E}(\vec{\mu}, \vec{\alpha}, f) > 0$, then

$$\frac{\mathcal{E}(\vec{\mu}, \vec{\alpha}, f)}{\mathcal{W}(\vec{\mu}, \vec{\alpha}, k)} \leq B.$$

To show this, let $\alpha > 1$ and $\beta \in (0,1)$ be arbitrary. Define

$$\epsilon_\alpha = \frac{\alpha - 1}{\alpha}\mathcal{E}(\vec{\mu}, \vec{\alpha}, f)$$
$$\epsilon_\beta = (1/\beta - 1)\mathcal{W}(\vec{\mu}, \vec{\alpha}, k)$$

and observe that $\epsilon_\alpha, \epsilon_\beta > 0$. Furthermore, for all $n = 1, \ldots, N$ choose a sequence $\vec{x}_n^{[M]} \in X^M$ such that $\vec{x}_n^{[M]} \xrightarrow{d_{\mathrm{KR}}} \mu_n$ for $M \to \infty$, and define $\vec{X}^{[M]} = \begin{pmatrix} \vec{x}_1^{[M]} & \cdots & \vec{x}_N^{[M]} \end{pmatrix}$. Choose $\ell_\epsilon \in \mathbb{N}_+$ such that for all $\ell \geq \ell_\epsilon$ we have

$$|\mathcal{E}(\vec{X}^{[M_\ell]}, \vec{\alpha}, f_{M_\ell}) - \mathcal{E}(\vec{\mu}, \vec{\alpha}, f)| \leq \epsilon_\alpha$$
$$|\mathcal{W}(\vec{X}^{[M_\ell]}, \vec{\alpha}, k_{M_\ell}) - \mathcal{W}(\vec{\mu}, \vec{\alpha}, k)| \leq \epsilon_\beta$$

and $\mathcal{W}(\vec{X}^{[M_\ell]}, \vec{\alpha}, k_{M_\ell}) > 0$. Such an $\ell_\epsilon$ exists because $k_M \xrightarrow{\mathcal{P}_1} k$ together with the continuity of $k_M$ and $k$ as well as the convergence of $\vec{x}_n^{[M]}$ to $\mu_n$ imply that $\mathcal{W}(\vec{X}^{[M_\ell]}, \vec{\alpha}, k_{M_\ell}) \to \mathcal{W}(\vec{\mu}, \vec{\alpha}, k)$, and $f_{M_\ell} \xrightarrow{\mathcal{P}_1} f$ together with the continuity of $f_M$ and $f$ imply that $\mathcal{E}(\vec{X}^{[M_\ell]}, \vec{\alpha}, f_{M_\ell}) \to \mathcal{E}(\vec{\mu}, \vec{\alpha}, f)$.

Let now $\ell \geq \ell_\epsilon$ be arbitrary. By definition of $\epsilon_\alpha$ we get $\alpha\epsilon_\alpha \leq (\alpha-1)\mathcal{E}(\vec{\mu}, \vec{\alpha}, f)$, which in turn leads to

$$
\begin{aligned}
\epsilon_\alpha &\leq \epsilon_\alpha - \alpha\epsilon_\alpha + (\alpha - 1)\mathcal{E}(\vec{\mu}, \vec{\alpha}, f) \\
&= -(\alpha - 1)\epsilon_\alpha + (\alpha - 1)\mathcal{E}(\vec{\mu}, \vec{\alpha}, f) \\
&= (\alpha - 1)(\mathcal{E}(\vec{\mu}, \vec{\alpha}, f) - \epsilon_\alpha) \\
&\leq (\alpha - 1)\mathcal{E}(\vec{X}^{[M_\ell]}, \vec{\alpha}, f_{M_\ell}),
\end{aligned}
$$

where we used in the last inequality that $\alpha - 1 > 0$ and by choice of $\ell_\epsilon$ we have $\mathcal{E}(\vec{\mu}, \vec{\alpha}, f) \leq \mathcal{E}(\vec{X}^{[M_\ell]}, \vec{\alpha}, f_{M_\ell}) + \epsilon_\alpha$. We can then continue with

$$
\begin{aligned}
\mathcal{E}(\vec{\mu}, \vec{\alpha}, f) &\leq \mathcal{E}(\vec{X}^{[M_\ell]}, \vec{\alpha}, f_{M_\ell}) + \epsilon_\alpha \\
&\leq \mathcal{E}(\vec{X}^{[M_\ell]}, \vec{\alpha}, f_{M_\ell}) + (\alpha - 1)\mathcal{E}(\vec{X}^{[M_\ell]}, \vec{\alpha}, f_{M_\ell}) \\
&= \alpha\mathcal{E}(\vec{X}^{[M_\ell]}, \vec{\alpha}, f_{M_\ell}).
\end{aligned}
$$

Next, by definition of $\epsilon_\beta$ and choice of $\ell_\epsilon$ we find that

$$
\begin{aligned}
\mathcal{W}(\vec{X}^{[M_\ell]}, \vec{\alpha}, k_{M_\ell}) &\leq \mathcal{W}(\vec{\mu}, \vec{\alpha}, k) + \epsilon_\beta \\
&= \mathcal{W}(\vec{\mu}, \vec{\alpha}, k) + (1/\beta - 1)\mathcal{W}(\vec{\mu}, \vec{\alpha}, k) \\
&= (1/\beta)\mathcal{W}(\vec{\mu}, \vec{\alpha}, k),
\end{aligned}
$$

hence

$$\frac{1}{\mathcal{W}(\vec{\mu}, \vec{\alpha}, k)} \leq \frac{1}{\beta\mathcal{W}(\vec{X}^{[M_\ell]}, \vec{\alpha}, k_{M_\ell})}.$$

Combining these results, we get that for all $\ell \geq \ell_\epsilon$

$$\frac{\mathcal{E}(\vec{\mu}, \vec{\alpha}, f)}{\mathcal{W}(\vec{\mu}, \vec{\alpha}, k)} \leq \frac{\alpha}{\beta}\frac{\mathcal{E}(\vec{X}^{[M_\ell]}, \vec{\alpha}, f_{M_\ell})}{\mathcal{W}(\vec{X}^{[M_\ell]}, \vec{\alpha}, k_{M_\ell})} \leq \frac{\alpha}{\beta}\mathcal{N}(f_{M_\ell}, k_{M_\ell}) = \frac{\alpha}{\beta}\|f_{M_\ell}\|_{M_\ell} \leq \frac{\alpha}{\beta}B.$$

Since $\alpha > 1$ and $\beta \in (0,1)$ were arbitrary, this shows that

$$\frac{\mathcal{E}(\vec{\mu}, \vec{\alpha}, f)}{\mathcal{W}(\vec{\mu}, \vec{\alpha}, k)} \leq B.$$

**Step 3** Let $(\vec{\mu}, \vec{\alpha}) \in \mathcal{P}(X)^N \times \mathbb{R}^N$ be arbitrary. If $\mathcal{W}(\vec{\mu}, \vec{\alpha}, k) = 0$, then we get from Step 1 that $\mathcal{E}(\vec{\mu}, \vec{\alpha}, f) = 0 \leq B$. Assume now $\mathcal{W}(\vec{\mu}, \vec{\alpha}, k) > 0$. If $\mathcal{E}(\vec{\mu}, \vec{\alpha}, f) = 0$, then again $\mathcal{E}(\vec{\mu}, \vec{\alpha}, f) = 0 \leq B$. If $\mathcal{E}(\vec{\mu}, \vec{\alpha}, f) > 0$, then Step 2 ensures that

$$\frac{\mathcal{E}(\vec{\mu}, \vec{\alpha}, f)}{\mathcal{W}(\vec{\mu}, \vec{\alpha}, k)} = \mathcal{D}(\vec{\mu}, \vec{\alpha}, f, k) \leq B.$$

Finally, if $\mathcal{E}(\vec{\mu}, \vec{\alpha}, f) < 0$, then again

$$\frac{\mathcal{E}(\vec{\mu}, \vec{\alpha}, f)}{\mathcal{W}(\vec{\mu}, \vec{\alpha}, k)} = \mathcal{D}(\vec{\mu}, \vec{\alpha}, f, k) < 0 \leq B.$$

Altogether, we get that $\mathcal{D}(\vec{\mu}, \vec{\alpha}, f, k) \leq B$. Since $(\vec{\mu}, \vec{\alpha})$ was arbitrary, maximization leads to $\mathcal{N}(f, k) \leq B < \infty$, hence $f \in H_k$ and $\|f\|_k = \mathcal{N}(f, k) \leq B$. $\qquad\square$

## A.2 Proofs for Section 3

In this section we provide the proofs for the results relating to approximation with kernels in the mean field limit.

*Proof. of Proposition 3.1* Let $f \in \mathcal{F}$ and $\epsilon > 0$ be arbitrary. Let $B \in \mathbb{R}_{\geq 0}$ and $f_M \in \mathcal{F}_M$, $\hat{f}_M \in H_M$, $M \in \mathbb{N}_+$, such that $f_M \xrightarrow{\mathcal{P}_1} f$, $\|f_M - \hat{f}_M\| \leq \frac{\epsilon}{5}$ and $\|\hat{f}_M\|_M \leq B$ for all $M \in \mathbb{N}_+$ (exist by definition of $\mathcal{F}$). Theorem 2.3 ensures that there exists a subsequence $(f_{M_\ell})_\ell$ and $\hat{f} \in H_k$ with $\|\hat{f}\|_k \leq B$ such that $\hat{f}_{M_\ell} \xrightarrow{\mathcal{P}_1} \hat{f}$ for $\ell \to \infty$. Choose now $L_1 \in \mathbb{N}_+$ such that for all $\ell \geq L_1$ we have

$$\sup_{\vec{x} \in X^{M_\ell}} |\hat{f}_{M_\ell}(\vec{x}) - \hat{f}(\hat{\mu}[\vec{x}])| \leq \frac{\epsilon}{5}$$

$$\sup_{\vec{x} \in X^{M_\ell}} |f_{M_\ell}(\vec{x}) - f(\hat{\mu}[\vec{x}])| \leq \frac{\epsilon}{5}.$$

Let now $\mu \in \mathcal{P}(X)$ be arbitrary and choose a sequence $\vec{x}_M \in X^M$ with $\hat{\mu}[\vec{x}_M] \xrightarrow{d_{\mathrm{KR}}} \mu$. Finally, let $L_2 \in \mathbb{N}_+$ such that for all $\ell \geq L_2$ we have

$$|f(\mu) - f(\hat{\mu}[\vec{x}_{M_\ell}])| \leq \frac{\epsilon}{5}$$

$$|\hat{f}(\mu) - \hat{f}(\hat{\mu}[\vec{x}_{M_\ell}])| \leq \frac{\epsilon}{5}$$

(such an $L_2$ exists due to the continuity of $f$ and $\hat{f}$).

We now have for $\ell \geq \max\{L_1, L_2\}$ that

$$|f(\mu) - \hat{f}(\mu)| \leq |f(\mu) - f(\hat{\mu}[\vec{x}_{M_\ell}])| + |f(\hat{\mu}[\vec{x}_{M_\ell}]) - f_{M_\ell}(\vec{x}_{M_\ell})| + |f_{M_\ell}(\vec{x}_{M_\ell}) - \hat{f}_{M_\ell}(\vec{x}_{M_\ell})|$$
$$+ |\hat{f}_{M_\ell}(\vec{x}_{M_\ell}) - \hat{f}(\hat{\mu}[\vec{x}_{M_\ell}])| + |\hat{f}(\hat{\mu}[\vec{x}_{M_\ell}]) - \hat{f}(\mu)|$$
$$\leq \frac{\epsilon}{5} + \frac{\epsilon}{5} + \frac{\epsilon}{5} + \frac{\epsilon}{5} + \frac{\epsilon}{5} = \epsilon.$$

Since $\mu$ was arbitrary, the result follows. $\qquad\square$

*Proof. of Remark 3.2* We first show that $\mathcal{F}$ is a subvectorspace. Let $f, g \in \mathcal{F}$ and $\lambda \in \mathbb{R}$, $\epsilon > 0$ be arbitrary. W.l.o.g. we can assume $\lambda \neq 0$. Choose sequences $f_M, g_M \in \mathcal{F}_M$, $\hat{f}_M, \hat{g}_M \in H_M$, $M \in \mathbb{N}_+$, and constants $B_f, B_g \in \mathbb{R}_{\geq 0}$ from the definition of $\mathcal{F}$ for $f$, $\frac{\epsilon}{2|\lambda|}$, and $g$, $\frac{\epsilon}{2}$, respectively. Let $M \in \mathbb{N}_+$, $\vec{x} \in X^M$ be arbitrary, then

$$|\lambda f_M(\vec{x}) + g(\vec{x}) - (\lambda f(\hat{\mu}[\vec{x}]) - g(\hat{\mu}[\vec{x}]))| \leq |\lambda||f_M(\vec{x}) - f(\hat{\mu}[\vec{x}])| + |g_M(\vec{x}) - g(\hat{\mu}[\vec{x}])|$$

together with $f_M \xrightarrow{\mathcal{P}_1} f$, $g_M \xrightarrow{\mathcal{P}_1} g$ shows that $\lambda f_M + g_M \xrightarrow{\mathcal{P}_1} \lambda f + g$.

Next, we have for all $M \in \mathbb{N}_+$ that

$$\|(\lambda f_M + g_M) - (\lambda \hat{f}_M + \hat{g}_M)\|_\infty \leq |\lambda|\|f_M - \hat{f}_M\|_\infty + \|g_M - \hat{g}_M\|_\infty \leq |\lambda|\frac{\epsilon}{2|\lambda|} + \frac{\epsilon}{2} = \epsilon.$$

Finally,

$$\|\lambda \hat{f}_M + \hat{g}_M\|_M \leq |\lambda|\|\hat{f}_M\|_M + \|\hat{g}_M\|_M \leq |\lambda|B_f + B_g,$$

establishing that $(\lambda \hat{f}_M + \hat{g}_M)_M$ is uniformly norm-bounded. Altogether, we have that $\lambda f + g \in \mathcal{F}$.

We now turn to the second claim. Let $(f^{(n)})_n \subseteq \mathcal{F}$ such that $f^{(n)} \to f$ for some $f \in C^0(\mathcal{P}(X), \mathbb{R})$ and for all $\bar{\epsilon} > 0$ there exist $f_M^{(n)} \in \mathcal{F}_M$, $\hat{f}_M^{(n)} \in H_M$, $(\rho_M)_M \subseteq \mathbb{R}_{\geq 0}$ and $B^{(n)} \in \mathbb{R}_{\geq 0}$ with $\rho_M \searrow 0$, $\|f_M^{(n)} - \hat{f}_M^{(n)}\|_\infty \leq \bar{\epsilon}$ and $\|\hat{f}_M^{(n)}\|_M \leq B^{(n)}$ for all $n, M \in \mathbb{N}_+$, and

$$\sup_{\vec{x} \in X^M} |f_M^{(n)}(\vec{x}) - f^{(n)}(\hat{\mu}[\vec{x}])| \leq \rho_M$$

for all $n, M \in \mathbb{N}_+$. We now show that $f \in \mathcal{F}$. For this, let $\epsilon > 0$ be arbitrary and choose $f_M^{(n)} \in \mathcal{F}_M$, $\hat{f}_M^{(n)} \in H_M$, $(\rho_M)_M \subseteq \mathbb{R}_{\geq 0}$ and $B^{(n)} \in \mathbb{R}_{\geq 0}$ as above with $\bar{\epsilon} = \frac{\epsilon}{4}$. Let $N \in \mathbb{N}_+$ be such that $\|f^{(m)} - f^{(n)}\|_\infty \leq \frac{\epsilon}{4}$ for all $m, n \geq N$ (such an $N$ exists since $(f^{(n)})_n$ converges in $C^0(\mathcal{P}(X), \mathbb{R})$ and hence is a Cauchy sequence). Furthermore, let $M_\rho \in \mathbb{N}_+$ be such that for all $M \geq M_\rho$ we have $\rho_M \leq \frac{\epsilon}{4}$. Define now $f_M = f_M^{(M)}$ and $\hat{f}_M = \hat{f}_M^{(M)}$ for $M = 1, \ldots, M_\rho - 1$, and $f_M = f_M^{(M+N)}$, $\hat{f}_M = \hat{f}_M^{(N)}$ for $M \geq M_\rho$.

**Step 1** Let $M \geq M_\rho$ and $\vec{x} \in X^M$ be arbitrary. We have

$$
\begin{aligned}
|f_M(\vec{x}) - f(\hat{\mu}[\vec{x}])| &= |f_M^{(N+M)}(\vec{x}) - f(\hat{\mu}[\vec{x}])| \\
&\leq |f_M^{(N+M)}(\vec{x}) - f^{(N+M)}(\hat{\mu}[\vec{x}])| + |f^{(N+M)}(\hat{\mu}[\vec{x}]) - f(\hat{\mu}[\vec{x}])| \\
&\leq \rho_M + \|f^{(N+M)} - f\|_\infty,
\end{aligned}
$$

and since the right hand side (which is independent of $\vec{x}$) converges to 0 for $M \to \infty$, we get $f_M \xrightarrow{\mathcal{P}_1} f$.

**Step 2** For $M = 1, \ldots, M_\rho$ we get

$$
\|f_M - \hat{f}_M\|_\infty = \|f_M^{(M)} - \hat{f}_M^{(M)}\|_\infty \leq \bar{\epsilon} \leq \epsilon.
$$

Let now $M \geq M_\rho$ and $\vec{x} \in X^M$ be arbitrary. We have

$$
\begin{aligned}
|f_M(\vec{x}) - \hat{f}_M(\vec{x})| &= |f_M^{(M+N)}(\vec{x}) - \hat{f}_M^{(N)}(\vec{x})| \\
&\leq |f_M^{(M+N)}(\vec{x}) - f^{(N+M)}(\hat{\mu}[\vec{x}])| + |f^{(N+M)}(\hat{\mu}[\vec{x}]) - f^{(N)}(\hat{\mu}[\vec{x}])| \\
&\quad + |f^{(N)}(\hat{\mu}[\vec{x}]) - f_M^{(N)}(\vec{x})| + |f_M^{(N)}(\vec{x}) - \hat{f}_M^{(N)}(\vec{x})| \\
&\leq \sup_{\vec{x}' \in X^M} |f_M^{(M+N)}(\vec{x}') - f^{(M+N)}(\hat{\mu}[\vec{x}'])| + \|f^{(M+N)} - f^{(N)}\|_\infty \\
&\quad + \sup_{\vec{x}' \in X^M} |f^{(N)}(\hat{\mu}[\vec{x}']) - f_M^{(N)}(\vec{x}')| + \|f_M^{(N)} - \hat{f}_M^{(N)}\|_\infty \\
&\leq \rho_M + \frac{\epsilon}{4} + \rho_M + \bar{\epsilon} \\
&\leq 4\frac{\epsilon}{4} = \epsilon,
\end{aligned}
$$

and since $\vec{x} \in X^M$ was arbitrary, we get $\|f_M - \hat{f}_M\|_\infty \leq \epsilon$.

**Step 3** For $M = 1, \ldots, M_\rho - 1$ we get by construction that $\|\hat{f}_M\|_M = \|\hat{f}_M^{(M)}\|_M \leq B^{(M)}$, and for $M \geq M_\rho$ we find $\|\hat{f}_M\|_M = \|\hat{f}_M^{(N)}\|_M \leq B^{(N)}$. Altogether, we get for $M \in \mathbb{N}_+$ that

$$
\|\hat{f}_M\|_M \leq \max\{B^{(1)}, \ldots, B^{(M_\rho - 1)}, B^{(N)}\}.
$$

Combining the three steps establishes that $f \in \mathcal{F}$. $\qquad\square$

Finally, here is the proof of the represnter theorem in the mean field limit.

*Proof. of Theorem 3.3* The existence and uniqueness of $f_M$ and $f$ follows from the well-known represrenter theorem (applied to all $k_M$ and $k$).

We now turn to the convergence of the minimizers. For all $M \in \mathbb{N}_+$ we have

$$
\lambda\|f_M^*\|_M \leq L(f_M^*(\vec{x}_1^{[M]}), \ldots, f_M^*(\vec{x}_N^{[M]})) + \lambda\|f\|_M \leq L(0, \ldots, 0),
$$

i.e., $\|f_M^*\|_M \leq L(0, \ldots, 0)/\lambda$. Define

$$
\begin{aligned}
\mathcal{L}_M : H_M \to \mathbb{R}_{\geq 0}, \ f \mapsto L(f(\vec{x}_1^{[M]}), \ldots, f(\vec{x}_N^{[M]})) + \lambda\|f\|_M \\
\mathcal{L} : H_k \to \mathbb{R}_{\geq 0}, \ f \mapsto L(f(\mu_1), \ldots, f(\mu_N)) + \lambda\|f\|_k,
\end{aligned}
$$

and let $f_M \in H_M$ with $f_M \xrightarrow{\mathcal{P}_1} f$ for some $f \in H_k$. The continuity of $f_M$, $f$ and $L$ as well as $\vec{x}_n^{[M]} \xrightarrow{d_{\text{KR}}} \mu_n$ for $M \to \infty$ and all $n = 1, \ldots, N$, imply then that

$\lim_{M\to\infty} L(f_M(\vec{x}_1^{[M]}), \ldots, f_M(\vec{x}_N^{[M]})) = L(f(\mu_1), \ldots, f(\mu_N))$. Combining this with Lemma 2.4 leads to

$$\mathcal{L}(f) \leq \liminf_{M\to\infty} \mathcal{L}_M(f).$$

Let now $f \in H_k$ be arbitrary and let $f_M \in H_M$ be the sequence from Lemma 2.5. Using the same arguments as above we find that

$$\limsup_{M\to\infty} \mathcal{L}_M(f_M) \leq \|f\|_k.$$

We have shown that $\mathcal{L}_M \xrightarrow{\Gamma} \mathcal{L}$ and hence Proposition B.3 ensures that there exists a subsequence $(f_{M_\ell}^*)_\ell$ such that $f_{M_\ell}^* \xrightarrow{\mathcal{P}_1} f^*$ and $\mathcal{L}_{M_\ell}(f_{M_\ell}^*) \to \mathcal{L}(f^*)$. $\qquad\square$

## A.3 Proofs for Section 4

*Proof. of Lemma 4.2* That $\ell$ is nonnegative is clear from the proof of Proposition 2.1. Let now all $\ell_M$ be convex and let $\mu \in \mathcal{P}(X)$, $y \in Y, t_1, t_2 \in \mathbb{R}$ and $\lambda \in (0,1)$ be arbitrary, and define $I = [\min\{t_1, t_2\}, \max\{t_1, t_2\}]$. Furthermore, let $\vec{x}_M \in X^M$ with $\vec{x}_M \xrightarrow{d_{\mathrm{KR}}} \mu$ for $M \to \infty$ and $\epsilon > 0$ be arbitrary. Choose now $M$ so large that

$$|\ell(\mu, y, \lambda t_1 + (1-\lambda)t_2) - \ell(\hat{\mu}[\vec{x}_M], y, \lambda t_1 + (1-\lambda)t_2)| \leq \frac{\epsilon}{6} \sup_{\substack{\vec{x}\in X^M \\ y'\in Y, t\in I}} |\ell_M(\vec{x}, y', t') - \ell(\hat{\mu}[\vec{x}], y', t')|$$

$$\leq \frac{\epsilon}{6}.$$

This is possible due to the continuity of $\ell$, as well as $\ell_M \xrightarrow{\mathcal{P}_1} \ell$. We then have

$$\ell(\mu, y, \lambda t_1 + (1-\lambda)t_2) \leq \ell(\hat{\mu}[\vec{x}], y, \lambda t_1 + (1-\lambda)t_2) + \frac{\epsilon}{6}$$

$$\leq \ell_M(\vec{x}_M, y, \lambda t_1 + (1-\lambda)t_2) + \frac{\epsilon}{3}$$

$$\leq \lambda \ell_M(\vec{x}_M, y, t_1) + (1-\lambda)\ell_M(\vec{x}_M, y, t_2) + \frac{\epsilon}{3}$$

$$\leq \lambda \ell(\hat{\mu}[\vec{x}_M], y, t_1) + (1-\lambda)\ell(\hat{\mu}[\vec{x}_M], y, t_2) + \frac{\epsilon}{3} + (\lambda + 1 - \lambda)\frac{\epsilon}{6}$$

$$\leq \lambda \ell(\mu, y, t_1) + (1-\lambda)\ell(\mu, y, t_2) + \epsilon,$$

and since $\epsilon > 0$ was arbitrary, this establishes

$$\ell(\mu, y, \lambda t_1 + (1-\lambda)t_2) \leq \lambda \ell(\mu, y, t_1) + (1-\lambda)\ell(\mu, y, t_2),$$

i.e., convexity of $\ell$. $\qquad\square$

*Proof. of Proposition 4.3* From Lemma 4.2 we get that $\ell$ is nonnegative and convex. The existence, uniqueness and the representation formulas follow then from the standard representer theorem, cf. e.g., [20, Theorem 5.5].

Furthermore, for all $M \in \mathbb{N}_+$ we have

$$\lambda\|f_{M,\lambda}^*\|_M^2 \leq \frac{1}{N}\sum_{n=1}^N \ell_M(\vec{x}_n^{[M]}, y_n^{[M]}, f_{M,\lambda}^*(\vec{x}_n^{[M]})) + \lambda\|f_{M,\lambda}^*\|_M^2$$

$$\leq \mathcal{R}_{\ell_M, D_N^{[M]}, \lambda}(0)$$

$$\leq NC_\ell,$$

hence $\|f_{M,\lambda}^*\|_M \leq \sqrt{\frac{NC_\ell}{\lambda}}$.

Let $f \in H_k$ and $(f_M)_M, f_M \in H_M$, such that $f_M \xrightarrow{\mathcal{P}_1} f$. From $D_N^{[M]} \xrightarrow{\mathcal{P}_1} D_N$ and the continuity of $\ell_M, \ell$, together with $\ell_M \xrightarrow{\mathcal{P}_1} \ell$ and the boundedness of $\{y_n^{[M]} \mid M \in \mathbb{N}_+, n = 1, \ldots, N\} \subseteq Y$ and $\{f_M(\vec{x}_n^{[M]}) \mid M \in \mathbb{N}_+, N = 1, \ldots, N\}$ we find that

$$\lim_M \frac{1}{N}\sum_{n=1}^N \ell_M(\vec{x}_n^{[M]}, y_n^{[M]}, f_M(\vec{x}_n^{[M]})) = \frac{1}{N}\sum_{n=1}^N \ell(\mu_n, y_n, f(\mu_n)).$$

Combining this with Lemma 2.4 and Lemma 2.5 then establishes that $\mathcal{R}_{\ell_M, D_N^{[M]}, \lambda} \xrightarrow{\Gamma} \mathcal{R}_{\ell, D_N, \lambda}$ and the remaining claims follow from Proposition B.3 and the uniqueness of the minimizers. □

*Proof. of Lemma 4.4* Let $\epsilon > 0$ be arbitrary. Recall from the proof of Proposition 4.3 that for all $M \in \mathbb{N}_+$ we have $\|f_{M,\lambda}^*\|_M \leq \sqrt{\frac{NC_\ell}{\lambda}}$, and hence for all $\vec{x} \in X^M$ we have

$$|f_{M,\lambda}^*(\vec{x})| \leq \|f_{M,\lambda}^*\|_k \|k_M(\cdot, \vec{x})\|_k$$
$$\leq \sqrt{\frac{NC_\ell}{\lambda}} \sqrt{C_k}.$$

A similar argument applies to $f_\lambda^* \in H_k$, so we can find a compact set $K \subseteq \mathbb{R}$ with

$$\{f_{M,\lambda}^*(\vec{x}_n^{[M]}) \mid M \in \mathbb{N}_+, n = 1, \ldots, N\} \cup \{f_\lambda^*(\mu_n) \mid n = 1, \ldots, N\} \subseteq K.$$

Choose now $m_\epsilon \in \mathbb{N}_+$ such that for all $m \geq m_\epsilon$ we have

$$\sup_{\substack{\vec{x} \in X^{M_m} \\ y \in Y}} |\ell_{M_m}(\vec{x}, y, f_{M_m,\lambda}^*(\vec{x})) - \ell_{M_m}(\vec{x}, y, f_\lambda^*(\hat{\mu}[\vec{x}]))| \leq \frac{\epsilon}{3}$$

$$\sup_{\substack{\vec{x} \in X^{M_m} \\ y \in Y, t \in K}} |\ell_{M_m}(\vec{x}, y, t) - \ell(\hat{\mu}[\vec{x}], y, t)| \leq \frac{\epsilon}{3}$$

$$\left| \int_{X^{M_m} \times Y} \ell(\hat{\mu}[\vec{x}], y, f_\lambda^*(\hat{\mu}[\vec{x}])) \mathrm{d}P^{[M_m]}(\vec{x}, y) - \int_{\mathcal{P}(X) \times Y} \ell(\mu, y, f_\lambda^*(\mu)) \mathrm{d}(\mu, y) \right| \leq \frac{\epsilon}{3}.$$

Such a $m_\epsilon$ exists since $f_{M_m,\lambda}^* \xrightarrow{\mathcal{P}_1} f_\lambda^*$ and all $\ell_{M_m}$ are uniformly Lipschitz continuous (first inequality), $\ell_{M_m} \xrightarrow{\mathcal{P}_1} \ell$ and $Y$ and $K$ are compact (second inequality), and $P^{[M]} \xrightarrow{\mathcal{P}_1} P$ as well as that $(\mu, y) \mapsto \ell(\mu, y, f_\lambda^*(\mu))$ is continuous and bounded (third inequality). We now have

$$\left| \mathcal{R}_{\ell_{M_m}, P^{[M_m]}}(f_{M_m,\lambda}^*) - \mathcal{R}_{\ell, P}(f_\lambda^*) \right|$$

$$\leq \left| \int_{X^{M_m} \times Y} \ell_{M_m}(\vec{x}, y, f_{M_m,\lambda}^*(\vec{x})) - \ell_{M_m}(\vec{x}, y, f_\lambda^*(\hat{\mu}[\vec{x}])) \mathrm{d}P^{[M_m]}(\vec{x}, y) \right|$$

$$+ \left| \int_{X^{M_m} \times Y} \ell_{M_m}(\vec{x}, y, f_\lambda^*(\hat{\mu}[\vec{x}])) - \ell(\hat{\mu}[\vec{x}], y, f_\lambda^*(\hat{\mu}[\vec{x}])) \mathrm{d}P^{[M_m]}(\vec{x}, y) \right|$$

$$+ \left| \int_{X^{M_m} \times Y} \ell(\hat{\mu}[\vec{x}], y, f_\lambda^*(\hat{\mu}[\vec{x}])) \mathrm{d}P^{[M_m]}(\vec{x}, y) - \int_{\mathcal{P}(X) \times Y} \ell(\mu, y, f_\lambda^*(\mu)) \mathrm{d}(\mu, y) \right|$$

$$\leq \int_{X^{M_m} \times Y} |\ell_{M_m}(\vec{x}, y, f_{M_m,\lambda}^*(\vec{x})) - \ell_{M_m}(\vec{x}, y, f_\lambda^*(\hat{\mu}[\vec{x}]))| \mathrm{d}P^{[M_m]}(\vec{x}, y)$$

$$+ \int_{X^{M_m} \times Y} |\ell_{M_m}(\vec{x}, y, f_\lambda^*(\hat{\mu}[\vec{x}])) - \ell(\hat{\mu}[\vec{x}], y, f_\lambda^*(\hat{\mu}[\vec{x}]))| \mathrm{d}P^{[M_m]}(\vec{x}, y)$$

$$+ \frac{\epsilon}{3}$$

$$\leq \epsilon,$$

and since $\epsilon > 0$ was arbitrary, the claim follows. □

*Proof. of Proposition 4.5* Observe that all $k_M$ are bounded measurable kernels, $\mathcal{R}_{\ell_M, P^{[M]}}(f_M) < \infty$ for all $f \in H_M$, $\ell_M$ is a convex, $P^{[M]}$-integrable Nemitskii loss (cf. Remark 4.1) and hence [20, Lemma 5.1, Theorem 5.2] guarantee the existence and uniqueness of $f_{M,\lambda}^*$. A completely analogous argument shows the existence and uniqueness of $f_\lambda^*$.

We now show that $\mathcal{R}_{\ell_M, P^{[M]}, \lambda} \xrightarrow{\Gamma} \mathcal{R}_{\ell, P, \lambda}$. For the $\Gamma$-$\liminf$-inequality, let $f_M \in H_M, f \in H_k$ be arbitrary with $f_M \xrightarrow{\mathcal{P}_1} f$, and let $\epsilon > 0$. Choose $M_\epsilon \in \mathbb{N}_+$ so large that for all $M \geq M_\epsilon$

$$\left| \int \ell(\hat{\mu}[\vec{x}], y, f(\hat{\mu}[\vec{x}]) \mathrm{d}P^{[M]}(\vec{x}, y)) - \int \ell(\mu, y, f(\mu)) \mathrm{d}P(\mu, y) \right| \leq \frac{\epsilon}{2}$$

(this is possible since $(\mu, y) \mapsto \ell(\mu, y, f(\mu))$ is bounded and continuous and $P^{[M]} \xrightarrow{\mathcal{P}_1} P$) and

$$\left| \ell_M(\vec{x}, y, f_M(\vec{x})) - \ell(\hat{\mu}[\vec{x}], y, f(\hat{\mu}[\vec{x}])) \right| \leq \frac{\epsilon}{2}$$

for all $\vec{x} \in X^M$, $y \in Y$ (this is possible due to the same argument used in the proof of Lemma 4.4). For $M \geq M_\epsilon$ we then find

$$
\begin{aligned}
\mathcal{R}_{\ell, P, \lambda}(f) &= \int \ell(\mu, y, f(\mu)) \mathrm{d}P(\mu, y) + \lambda \|f\|_k^2 \\
&\leq \int \ell_M(\vec{x}, y, f_M(\vec{x})) \mathrm{d}P^{[M]}(\vec{x}, y) \\
&\quad + \left| \int \ell(\hat{\mu}[\vec{x}], y, f(\hat{\mu}[\vec{x}]) \mathrm{d}P^{[M]}(\vec{x}, y)) - \int \ell(\mu, y, f(\mu)) \mathrm{d}P(\mu, y) \right| \\
&\quad + \left| \int \ell_M(\vec{x}, y, f_M(\vec{x})) - \ell(\hat{\mu}[\vec{x}], y, f(\hat{\mu}[\vec{x}])) \mathrm{d}P^{[M]}(\vec{x}, y) \right| + \lambda \|f\|_k^2 \\
&\leq \int \ell_M(\vec{x}, y, f_M(\vec{x})) \mathrm{d}P^{[M]}(\vec{x}, y) + \lambda \liminf_M \|f_M\|_M^2 + \epsilon,
\end{aligned}
$$

where we used Lemma 2.4 in the last inequality.

For the $\Gamma$-lim sup-inequality, let $f \in H_k$ be arbitrary and let $(f_M)_M$ be the recovery sequence from Lemma 2.5. The desired inequality then follows by repeating the arguments from above.

Finally, using exactly the same argument as in the proof of Proposition 4.3 shows that $\|f_{M,\lambda}^*\|_M \leq \sqrt{\frac{NC_\ell}{\lambda}}$, so we can apply Proposition B.3 and the result follows. $\qquad \square$

*Proof. of Proposition 4.7* Let $(\epsilon_n)_n \subseteq \mathbb{R}_{>0}$ with $\epsilon_m \searrow 0$. We construct a strictly increasing sequence $(M_n)_n$ such that

$$\left| \mathcal{R}_{\ell_{M_n}, P^{[M_n]}}^{H_{M_n}*} - \mathcal{R}_{\ell, P}^{H_k*} \right| \leq \epsilon_n$$

for all $n \in \mathbb{N}_+$.

We start with $n = 1$: Since $A_2(0) = 0$ and $A_2$ is continuous in 0, cf. [20, Lemma 5.15], there exists $\lambda_1' \in \mathbb{R}_{>0}$ such that $A_2(\lambda) \leq \frac{\epsilon_1}{3}$ for all $0 < \lambda \leq \lambda_1'$. From Assumption 4.6 we get $\lambda_1'' \in \mathbb{R}_{>0}$ such that for all $M \in \mathbb{N}_+$ we have $A_2^{[M]}(\lambda) \leq \frac{\epsilon_1}{3}$ for all $0 < \lambda \leq \lambda_1''$. Define now $\lambda_1 = \min\{\lambda_1', \lambda_1''\}$, and observe that $\lambda_1 > 0$. Proposition 4.5 ensures the existence of a strictly increasing sequence $(M_m^{(1)})_m \subseteq \mathbb{N}_+$ with

$$\mathcal{R}_{\ell_{M_m^{(1)}}, P^{[M_m^{(1)}]}, \lambda_1}^{H_{M_m^{(1)}}*} \to \mathcal{R}_{\ell, P, \lambda_1}^{H_k*}$$

for $m \to \infty$. Choose $m_1 \in \mathbb{N}_+$ such that for all $m \geq m_1$ we have

$$\left| \mathcal{R}_{\ell_{M_m^{(1)}}, P^{[M_m^{(1)}]}, \lambda_1}^{H_{M_m^{(1)}}*} - \mathcal{R}_{\ell, P, \lambda_1}^{H_k*} \right| \leq \frac{\epsilon_1}{3}.$$

We now set $M_1 = M_{m_1}^{(1)}$ and get that

$$
\begin{aligned}
\left| \mathcal{R}_{\ell_{M_1}, P^{[M_1]}}^{H_{M_1}*} - \mathcal{R}_{\ell, P}^{H_k*} \right| &\leq \left| \mathcal{R}_{\ell_{M_{m_1}^{(1)}}, P^{[M_{m_1}^{(1)}]}}^{H_{M_{m_1}^{(1)}}*} - \mathcal{R}_{\ell_{M_{m_1}^{(1)}}, P^{[M_{m_1}^{(1)}]}, \lambda_1}^{H_{M_{m_1}^{(1)}}*} \right| + \left| \mathcal{R}_{\ell_{M_{m_1}^{(1)}}, P^{[M_{m_1}^{(1)}]}, \lambda_1}^{H_{M_{m_1}^{(1)}}*} - \mathcal{R}_{\ell, P, \lambda_1}^{H_k*} \right| \\
&\quad + \left| \mathcal{R}_{\ell, P, \lambda_1}^{H_k*} - \mathcal{R}_{\ell, P}^{H_k*} \right| \\
&\leq A_2^{[M_m^{(1)}]}(\lambda_1) + \frac{\epsilon_1}{3} + A_2(\lambda_1) \\
&\leq \epsilon_1.
\end{aligned}
$$

We can now repeat the argument from above inductively: Suppose we have constructed our subsequence up to $n \in \mathbb{N}_+$, i.e., $M_1, \ldots, M_n$. Choose $\lambda' \in \mathbb{R}_{>0}$ such that $A_2(\lambda) \leq \frac{\epsilon_{n+1}}{3}$ for

all $0 < \lambda \leq \lambda'$ (exists due to continuity), and $\lambda'' \in \mathbb{R}_{>0}$ such that for all $M \in \mathbb{N}_+$ we have $A_2^{[M]}(\lambda) \leq \frac{\epsilon_{n+1}}{3}$ for all $0 < \lambda \leq \lambda''$ (using Assumption 4.6). Define now $\lambda_{n+1} = \min\{\lambda', \lambda''\}$, and observe that $\lambda_{n+1} > 0$. Proposition 4.5 ensures the existence of a strictly increasing sequence $\left(M_m^{(n+1)}\right)_m$ such that

$$\mathcal{R}^{H_{M_m^{(n+1)}}*}_{\ell_{M_m^{(n+1)}},P[M_m^{(n+1)}],\lambda_{n+1}} \to \mathcal{R}^{H_k*}_{\ell,P,\lambda_{n+1}}$$

for $m \to \infty$. Choose $m_{n+1}$ such that for all $m \geq m_{n+1}$ we have

$$\left| \mathcal{R}^{H_{M_m^{(n+1)}}*}_{\ell_{M_m^{(n+1)}},P[M_m^{(n+1)}],\lambda_{n+1}} - \mathcal{R}^{H_k*}_{\ell,P,\lambda_{n+1}} \right| \leq \frac{\epsilon_{n+1}}{3}.$$

Define now $M_{n+1} = \max\{M_n + 1, M_{m_{n+1}}^{(n+1)}\}$, then we get

$$\left| \mathcal{R}^{H_{M_{n+1}}*}_{\ell_{M_{n+1}},P[M_{n+1}]} - \mathcal{R}^{H_k*}_{\ell,P} \right| \leq \left| \mathcal{R}^{H_{M_{m_{n+1}}^{(n+1)}}*}_{\ell_{M_{m_{n+1}}^{(n+1)}},P[M_{m_{n+1}}^{(n+1)}]} - \mathcal{R}^{H_{M_{m_{n+1}}^{(n+1)}}*}_{\ell_{M_{m_{n+1}}^{(n+1)}},P[M_{m_{n+1}}^{(n+1)}],\lambda_{n+1}} \right|$$

$$+ \left| \mathcal{R}^{H_{M_{m_{n+1}}^{(n+1)}}*}_{\ell_{M_{m_{n+1}}^{(n+1)}},P[M_{m_{n+1}}^{(n+1)}],\lambda_{n+1}} - \mathcal{R}^{H_k*}_{\ell,P,\lambda_{n+1}} \right|$$

$$+ \left| \mathcal{R}^{H_k*}_{\ell,P,\lambda_{n+1}} - \mathcal{R}^{H_k*}_{\ell,P} \right|$$

$$\leq A_2^{M_{m_{n+1}}^{(n+1)}}(\lambda_{n+1}) + \frac{\epsilon_{n+1}}{3} + A_2(\lambda_{n+1})$$

$$\leq \epsilon_{n+1}.$$

The resulting sequence $(M_n)_n$ fulfills then

$$\mathcal{R}^{H_{M_n}*}_{\ell_{M_n},P[M_n]} \to \mathcal{R}^{H_k*}_{\ell,P}$$

for $n \to \infty$. $\qquad\square$

# B    Additional technical results

In this section we state and prove two technical results that play an important role in the proofs of the main results.

## B.1    A characterization of RKHS functions

Here we recall the following characterization of RKHS functions from [31, Section I.4]. Let $\mathcal{X} \neq \emptyset$ be arbitrary. For $k : \mathcal{X} \times \mathcal{X} \to \mathbb{R}$ symmetric and positive semidefinite and some $f \in \mathbb{R}^{\mathcal{X}}$ as well as $N \in \mathbb{N}_+, \vec{x} \in \mathcal{X}^N, \vec{\alpha} \in \mathbb{R}^N$ define

$$\mathcal{E}(\vec{x}, \vec{\alpha}, f) = \sum_{n=1}^{N} \alpha_n f(x_n)$$

$$\mathcal{W}(\vec{x}, \vec{\alpha}, k) = \sqrt{\sum_{i,j=1}^{N} \alpha_i \alpha_j k(x_j, x_i)},$$

where we might omit some arguments if they are clear. Furthermore, define

$$\mathcal{D}(\vec{x}, \vec{\alpha}, f, k) = \begin{cases} \frac{\mathcal{E}(\vec{x},\vec{\alpha},f)}{\mathcal{W}(\vec{x},\vec{\alpha},k)} & \text{if } \mathcal{E}(\vec{x}, \vec{\alpha}, f) \neq 0, \mathcal{W}(\vec{x}, \vec{\alpha}, k) \neq 0 \\ 0 & \text{if } \mathcal{E}(\vec{x}, \vec{\alpha}, f) = \mathcal{W}(\vec{x}, \vec{\alpha}, k) = 0 \\ \infty & \text{if } \mathcal{E}(\vec{x}, \vec{\alpha}, f) \neq 0, \mathcal{W}(\vec{x}, \vec{\alpha}, k) = 0 \end{cases}$$

and

$$\mathcal{N}(f,k) = \sup_{\substack{(\vec{x},\vec{\alpha})\in\mathcal{X}^N\times\mathbb{R}^N \\ N\in\mathbb{N}_+}} \mathcal{D}(\vec{x},\vec{\alpha},f,k).$$

We collect now some simple facts that will be used repeatedly.

Let $\vec{x}\in\mathcal{X}^N$, $\vec{\alpha}\in\mathbb{R}^N$, $N\in\mathbb{N}_+$, be arbitrary, and define

$$f = \sum_{n=1}^N \alpha_n k(\cdot,x_n) \in H_k^{\text{pre}}.$$

1. By construction, $\mathcal{W}(\vec{x},\vec{\alpha},k) \in \mathbb{R}_{\geq 0}$ (recall that $k$ is positive semidefinite).

2. Since $f \in H_k^{\text{pre}}$, its RKHS norm has an explicit form and we find

$$\|f\|_k = \sqrt{\sum_{i,j=1}^N \alpha_i\alpha_j k(x_j,x_i)} = \mathcal{W}(\vec{x},\vec{\alpha},k).$$

This also implies that $f \equiv 0$ if and only if $\mathcal{W}(\vec{x},\vec{\alpha},k) = 0$.

3. If $\mathcal{W}(\vec{x},\vec{\alpha},k) > 0$, then

$$\begin{aligned}
\mathcal{D}(\vec{x},\vec{\alpha},f,k) &= \frac{\mathcal{E}(\vec{x},\vec{\alpha},f)}{\mathcal{W}(\vec{x},\vec{\alpha},k)} \\
&= \frac{\sum_{i=1}^N \alpha_i f(x_i)}{\sqrt{\sum_{i,j=1}^N \alpha_i\alpha_j k(x_j,x_i)}} \\
&= \frac{\sum_{i,j=1}^N \alpha_i\alpha_j k(x_j,x_i)}{\sqrt{\sum_{i,j=1}^N \alpha_i\alpha_j k(x_j,x_i)}} \\
&= \frac{\mathcal{W}(\vec{x},\vec{\alpha},k)^2}{\mathcal{W}(\vec{x},\vec{\alpha},k)} = \mathcal{W}(\vec{x},\vec{\alpha},k).
\end{aligned}$$

We can now state the characterization result.

**Theorem B.1.** Let $k : \mathcal{X}\times\mathcal{X}\to\mathbb{R}$ be a kernel and $f\in\mathbb{R}^{\mathcal{X}}$. Then $f\in H_k$ if and only if $\mathcal{N}(f,k) < \infty$. If $f\in H_k$, then $\|f\|_k = \mathcal{N}(f,k)$.

For convenience, we provide a full self-contained proof of this result.

*Proof.* **Step 1** First, we show that for $f\in H_k$, we have $\|f\|_k = \mathcal{N}(f,k)$.

$\mathcal{N}(f,k) \leq \|f\|_k$: Let $N\in\mathbb{N}_+$ and $(\vec{x},\vec{\alpha})\in\mathcal{X}^N\times\mathbb{R}^N$ be arbitrary. Observe that

$$\begin{aligned}
\mathcal{E}(\vec{x},\vec{\alpha},f) &= \sum_{n=1}^N \alpha_n f(x_n) \\
&= \sum_{n=1}^N \alpha_n \langle f, k(\cdot,x_n)\rangle_k \\
&= \langle f, \sum_{n=1}^N \alpha_n k(\cdot,x_n)\rangle_k \\
&\leq \|f\|_k \left\|\sum_{n=1}^N \alpha_n k(\cdot,x_n)\right\|_k \\
&= \|f\|_k \mathcal{W}(\vec{x},\vec{\alpha},k).
\end{aligned}$$

If $\mathcal{W}(\vec{x},\vec{\alpha},k) = \|\sum_{n=1}^N \alpha_n k(\cdot,x_n)\|_k = 0$, then $\sum_{n=1}^N \alpha_n k(\cdot,x_n) = 0_{H_k}$, hence $\mathcal{E}(\vec{x},\vec{\alpha},f) = \langle f, 0_{H_k}\rangle_k = 0$ and by definition $\mathcal{D}(\vec{x},\vec{\alpha},f,k) = 0 \leq \|f\|_k$.

If $\mathcal{W}(\vec{x}, \vec{\alpha}, k) > 0$, we can rearrange to get

$$\frac{\mathcal{E}(\vec{x}, \vec{\alpha}, f)}{\mathcal{W}(\vec{x}, \vec{\alpha}, k)} = \mathcal{D}(\vec{x}, \vec{\alpha}, f, k) \leq \|f\|_k.$$

Since $(\vec{x}, \vec{\alpha})$ was arbitrary, we find that $\mathcal{N}(\vec{x}, \vec{\alpha}, f, k) \leq \|f\|_k$.

$\mathcal{N}(f, k) \geq \|f\|_k$: Let $\epsilon > 0$ and choose $f_\epsilon = \sum_{n=1}^{N} \alpha_n k(\cdot, x_n) \in H_k^{\text{pre}}$ such that $\|f - f_\epsilon\|_k < \epsilon$. If $\mathcal{W}(\vec{x}, \vec{\alpha}, k) = \|f_\epsilon\|_k = 0$, then $f_\epsilon = 0_{H_k}$ and hence $\mathcal{E}(\vec{x}, \vec{\alpha}, f) = \langle f, f_\epsilon \rangle_k = \langle f, 0_{H_k} \rangle_k = 0$. By definition, this then shows

$$\mathcal{D}(\vec{x}, \vec{\alpha}, f) = 0 = \|f_\epsilon\|_k \geq \|f\|_k - \epsilon.$$

Before we continue, note that for all $f_1, f_2 \in H_k$ we have

$$|\mathcal{E}(\vec{x}, \vec{\alpha}, f_1) - \mathcal{E}(\vec{x}, \vec{\alpha}, f_2)| = \left| \sum_{n=1}^{N} \alpha_n (f_1(x_n) - f_2(x_n)) \right|$$

$$= \left| \sum_{n=1}^{N} \alpha_n \langle f_1 - f_2, k(\cdot, x_n) \rangle_k \right|$$

$$= \left| \langle f_1 - f_2, \sum_{n=1}^{N} \alpha_n k(\cdot, x_n) \rangle_k \right|$$

$$\leq \|f_1 - f_2\|_k \|f_\epsilon\|_k.$$

Assume now that $\mathcal{W}(\vec{x}, \vec{\alpha}, k) > 0$, then we get

$$\mathcal{D}(\vec{x}, \vec{\alpha}, f, k) = \frac{\mathcal{E}(\vec{x}, \vec{\alpha}, f)}{\mathcal{W}(\vec{x}, \vec{\alpha}, k)}$$

$$\geq \frac{\mathcal{E}(\vec{x}, \vec{\alpha}, f_\epsilon)}{\mathcal{W}(\vec{x}, \vec{\alpha}, k)} - \frac{\|f - f_\epsilon\|_k \|f_\epsilon\|_k}{\mathcal{W}(\vec{x}, \vec{\alpha}, k)}$$

$$\geq \frac{\mathcal{E}(\vec{x}, \vec{\alpha}, f_\epsilon)}{\mathcal{W}(\vec{x}, \vec{\alpha}, k)} - \frac{\epsilon \|f_\epsilon\|_k}{\mathcal{W}(\vec{x}, \vec{\alpha}, k)}$$

$$= \mathcal{W}(\vec{x}, \vec{\alpha}, k) - \epsilon$$

$$= \|f_\epsilon\|_k - \epsilon$$

$$\geq \|f\|_k - 2\epsilon$$

Altogether, by definition of $\mathcal{N}(f, k)$, we get that

$$\mathcal{N}(f, k) \geq \mathcal{D}(\vec{x}, \vec{\alpha}, f, k) \geq \|f\|_k - 2\epsilon.$$

Since $\epsilon > 0$ was arbitrary, we find that $\mathcal{N}(f, k) \geq \|f\|_k$.

**Step 2** Let $f \in \mathbb{R}^{\mathcal{X}}$ be arbitrary. We show that if $\mathcal{N}(f, k) < \infty$, then

$$\ell_f : H_k^{\text{pre}} \to \mathbb{R}$$

$$\sum_{n=1}^{N} \alpha_n k(\cdot, x_n) \mapsto \sum_{n=1}^{N} \alpha_n f(x_n)$$

is a well-defined, linear and continuous (w.r.t. $\| \cdot \|_k$) map.

To establish the *well-posedness*, let $(\vec{x}, \vec{\alpha}) \in \mathcal{X}^N \times \mathbb{R}^N$ and $(\vec{y}, \vec{\beta}) \in \mathcal{X}^M \times \mathbb{R}^M$ such that

$$\sum_{n=1}^{N} \alpha_n k(\cdot, x_n) = \sum_{m=1}^{M} \beta_m k(\cdot, y_m) \in H_k^{\text{pre}}.$$

This implies that

$$\sum_{n=1}^{N} \alpha_n k(\cdot, x_n) + \sum_{m=1}^{M} (-\beta_m) k(\cdot, y_m) = 0_{H_k}$$

and hence $\mathcal{W}((\vec{x}, \vec{y}), (\vec{\alpha}, -\vec{\beta}), k) = \| \sum_{n=1}^{N} \alpha_n k(\cdot, x_n) + \sum_{m=1}^{M} (-\beta_m) k(\cdot, y_m) \|_k = 0$. Assume now that

$$\sum_{n=1}^{N} \alpha_n f(x_n) \neq \sum_{m=1}^{m} \beta_m f(x_m),$$

then we get that

$$\sum_{n=1}^{N} \alpha_n f(x_n) + \sum_{m=1}^{m} (-\beta_m) f(x_m) = \mathcal{E}((\vec{x}, \vec{y}), (\vec{\alpha}, -\vec{\beta}), f) \neq 0$$

which by definition implies that $\mathcal{D}((\vec{x}, \vec{y}), (\vec{\alpha}, -\vec{\beta}), f, k) = \infty$ and therefore $\mathcal{N}(f, k) = \infty$, a contradiction.

The *linearity* is then clear. Finally, to show the *continuity*, let $H_k^{\text{pre}} \ni f_0 = \sum_{n=1}^{N} \alpha_n k(\cdot, x_n)$ be arbitrary and set $\vec{x} = (x_1 \quad \cdots \quad x_N)$, $\vec{\alpha} = (\alpha_1 \quad \cdots \quad \alpha_N)$, then

$$
\begin{aligned}
|\ell_f(f_0)| &= \left| \sum_{n=1}^{N} \alpha_n f(x_n) \right| \\
&= |\mathcal{E}(\vec{x}, \vec{\alpha}, f)| \\
&\leq \mathcal{N}(f, k) \mathcal{W}(\vec{x}, \vec{\alpha}, k) \\
&= \mathcal{N}(f, k) \| f_0 \|_k.
\end{aligned}
$$

Since $\mathcal{N}(f, k)$ is finite and independent of $f_0$, and $\ell_f$ is a linear map, this shows the continuity of $\ell_f$.

**Step 3** Let $f \in \mathbb{R}^{\mathcal{X}}$ such that $\mathcal{N}(f, k) < \infty$. Since according to Step 2 $\ell_f$ is a linear and continuous map on $H_k^{\text{pre}}$ and the latter is dense in $H_k$, there exists a unique linear and continuous extension $\bar{\ell}_f : H_k \to \mathbb{R}$ of $\ell_f$. Furthermore, from the Riesz Representation Theorem there exists a unique $\hat{f} \in H_k$ with $\bar{\ell}_f = \langle \cdot, \hat{f} \rangle_k$. For all $x \in \mathcal{X}$ we then get

$$
\begin{aligned}
\hat{f}(x) &= \langle \hat{f}, k(\cdot, x) \rangle_k \\
&= \langle k(\cdot, x), \hat{f} \rangle_k \\
&= \bar{\ell}_f(k(\cdot, x)) \\
&= \ell_f(k(\cdot, x)) \\
&= f(x),
\end{aligned}
$$

hence $f = \hat{f} \in H_k$. $\qquad \square$

## B.2 A $\Gamma$-convergence argument

We use repeatedly the concept of $\Gamma$-convergence, see for example [32]. For convenience, in this section we summarize the well-known and standard main argument, roughly following [33, Chapter 5].

**Definition B.2.** Let $F_M : H_M \to \mathbb{R} \cup \{\infty\}$ and $F : H_k \to \mathbb{R} \cup \{\infty\}$. We say that $F_M$ $\Gamma$-converges to $F$ and write $F_M \xrightarrow{\Gamma} F$, if

1. For all sequences $(f_M)_M$, $f_M \in H_M$, with $f_M \xrightarrow{\mathcal{P}_1} f$ for some $f \in H_k$, we have
$$F(f) \leq \liminf_M F_M(f_M).$$

2. For all $f \in H_k$ there exists a sequence $(f_M)_M$ with $f_M \in H_M$ such that $f_M \xrightarrow{\mathcal{P}_1} f$ and
$$F(f) \geq \limsup_M F_M(f_M).$$

The sequence in the second item is commonly called a *recovery sequence* (for $f$).

**Proposition B.3.** Let $F_M \xrightarrow{\Gamma} F$ and $f_M^* \in \operatorname{argmin}_{f \in H_M} F_M(f)$ for all $M \in \mathbb{N}$ (in particular, all the minima are attained). If there exists $B \in \mathbb{R}_{\geq 0}$ such that $\| f_M^* \|_M \leq B$ for all $M \in \mathbb{N}$, then there exists a subsequence $(f_{M_\ell}^*)_\ell$ and $f^* \in H_k$ such that $f_{M_\ell}^* \xrightarrow{\mathcal{P}_1} f^*$. Furthermore, $F_{M_\ell}(f_{M_\ell}^*) \to F(f^*)$.

*Proof.* From Theorem 2.3 we get the existence of $(f^*_{M_\ell})_\ell$ and $f^* \in H_k$, and that $f^*_{M_\ell} \xrightarrow{\mathcal{P}_1} f^*$. Let $f \in H_k$ be arbitrary and let $(f_M)_M$ be a recovery sequence for $f$. We then have

$$
\begin{aligned}
F(f) &\geq \limsup_{M} F_M(f_M) \\
&\geq \limsup_{M_\ell} F_{M_\ell}(f_{M_\ell}) \\
&\geq \liminf_{M_\ell} F_{M_\ell}(f_{M_\ell}) \\
&\geq \liminf_{M_\ell} F_{M_\ell}(f^*_{M_\ell}) \\
&\geq F(f^*),
\end{aligned}
$$

where we used the $\limsup$-inequality of $\Gamma$-convergence in the first step, standard properties of $\limsup$ and $\liminf$ in the second and third step, the fact that $f^*_{M_\ell}$ is a minimizer of $F_{M_\ell}$ in the fourth step, and finally the $\liminf$-inequality of $\Gamma$-convergence. Since $f \in H_k$ was arbitrary, this shows that $f^*$ is a minimizer of $F$.

Furthermore, let $(f_M)_M$ be a recovery sequence for $f^*$, then

$$
\begin{aligned}
F(f^*) &\geq \limsup_{M} F_M(f_M) \\
&\geq \limsup_{\ell} F_{M_\ell}(f_{M_\ell}) \\
&\geq \limsup_{\ell} F_{M_\ell}(f^*_{M_\ell}),
\end{aligned}
$$

where we used the $\limsup$-inequality in the first step, an elementary property of $\limsup$ in the second step, and finally that $f^*_{M_\ell}$ is a minimizer of $F_{M_\ell}$. Since $f^*_{M_\ell} \xrightarrow{\mathcal{P}_1} f^*$, the $\liminf$-inequality of $\Gamma$-convergence implies that

$$
F(f^*) \leq \liminf_{\ell} F_{M_\ell}(f^*_{M_\ell}),
$$

so we find that

$$
\limsup_{\ell} F_{M_\ell}(f^*_{M_\ell}) \leq F(f^*) \leq \liminf_{\ell} F_{M_\ell}(f^*_{M_\ell}),
$$

establishing that $F_{M_\ell}(f^*_{M_\ell}) \to F(f^*)$. $\qquad\square$

