# On kernel-based statistical learning theory in the mean field limit

## Abstract

In many applications of machine learning, a large number of variables are considered. Motivated by machine learning of interacting particle systems, we consider the situation when the number of input variables goes to infinity. First, we continue the recent investigation of the mean field limit of kernels and their reproducing kernel Hilbert spaces, completing the existing theory. Next, we provide results relevant for approximation with such kernels in the mean field limit, including a representer theorem. Finally, we use these kernels in the context of statistical learning in the mean field limit, focusing on Support Vector Machines. In particular, we show mean field convergence of empirical and infinite-sample solutions as well as the convergence of the corresponding risks. On the one hand, our results establish rigorous mean field limits in the context of kernel methods, providing new theoretical tools and insights for large-scale problems. On the other hand, our setting corresponds to a new form of limit of learning problems, which seems to have not been investigated yet in the statistical learning theory literature.

## 1 Introduction

Models with many variables play an important role in many fields of mathematical and physical sciences. In this context, going to the limit of infinitely many variables is an important analysis and modeling approach. A classic example are interacting particle systems; these are usually modeled as dynamical systems describing the temporal evolution of many interacting objects. In physics, such systems were first investigated in the context of gas dynamics, cf. [11]. Since even small volumes of gases typically contain an enormous number of molecules, a microscopic modeling approach quickly becomes infeasible and one considers the evolution of densities instead [12]. In the past decades, interacting particle systems arising from many different domains have been considered, for example, animal movement [4, 23], social and political dynamics [31, 10], crowd modeling and control [17, 15, 1], swarms of robots [28, 27, 13] or vehicular traffic [32]. There is now a vast literature on such applications, and we refer to the surveys [26, 33, 21] as starting points. A prototypical example of such a system is given by $\dot{x}_i = \frac{1}{M} \sum_{j=1}^{M} \phi(x_i, x_j)(x_j - x_i)$, for $i = 1, \ldots, M$, where $M \in \mathbb{N}_+$ particles or agents are modelled by their state $x_i \in \mathbb{R}^d$, $i = 1, \ldots, M$, evolving according to some interaction rule $\phi : \mathbb{R}^d \times \mathbb{R}^d \to \mathbb{R}$. Typical questions then concern the long-term behavior of such systems, in particular, emergent phenomena like consensus or alignment [9]. While first-principles modeling has been very successful for interacting particle systems in physical domains, using this approach to model the interaction rules in complex domains like social and opinion dynamics, pedestrian and animal movement or vehicular traffic, can be problematic. Therefore, learning interaction rules from data has been recently intensively investigated, for example, in the pioneering works [6, 25]. The data consists typically of (sampled) trajectories of the particle states, potentially with measurement noise, and the goal is to learn a good approximation of the interaction rule $\phi$.

Submitted to 37th Conference on Neural Information Processing Systems (NeurIPS 2023). Do not distribute.

A related question is that of learning a function $F_M : (\mathbb{R}^d)^M \to \mathbb{R}$ of the particle states. This corresponds to a (real-valued) feature of a given population, which depends on each individual particle state. Similar to the case of the interaction rule, we might not be able to model such a feature, but we could measure it at different time instants and try to learn this