# OpenReview forum: "On kernel-based statistical learning theory in the mean field limit"
_NeurIPS.cc/2023/Conference — NeurIPS 2023 poster_

### Official Review · Reviewer_Vo6s · 2023-07-05

**Soundness:** 3 good
**Presentation:** 3 good
**Contribution:** 2 fair
**Rating:** 6
**Confidence:** 3

**Summary:**

The authors explore the mean field limit of kernels and their reproducing kernel Hilbert spaces, providing novel theoretical tools and insights for tackling large-scale problems. Furthermore, they employ these kernels in statistical learning, with a particular focus on Support Vector Machines. This new form of limit for learning problems has not been investigated before in the statistical learning theory literature, making the authors' research a valuable contribution to the field.

**Strengths:**

1. The authors provide a comprehensive investigation of the mean field limit of kernels and their reproducing kernel Hilbert spaces.

2. The research extends the basic theory of mean field kernels, providing novel approximation results and a variant of the representer theorem for mean field kernels.

3. The authors introduce a new form of limit for learning problems and provide convergence results for Support Vector Machines using mean field kernels.

**Weaknesses:**

1. The limitations or drawbacks of the research are not explicitly stated.

2. Section 4 only provides a consistency result instead of a generalization error bound.

**Questions:**

1. What is the main improvement of this paper compared to the reference [15]?

2. What is the relationship and difference of this study compared to neural tangent kernel? What is the advantage of mean field kernels?

3. Any concrete examples for mean field kernels?

**Limitations:**

Yes

---

> ### Author Rebuttal · Authors · 2023-08-08
>
> Thank you very much for your review and insightful questions. Below we address your concerns and answer the posed questions.
>
> **Weaknesses**
> >Section 4 only provides a consistency result instead of a generalization error bound.
>
> Indeed, we only consider convergence per se in Section 4 and do not provide rates or generalization bounds. These latter aspects are very interesting and relevant, and subject ongoing investigations. However, we would argue that establishing consistency is already interesting and challenging, which motivated us to focus on this in the present work.
>
> **Questions**
> >What is the main improvement of this paper compared to the reference [15]?
>
> Reference [15] does not consider mean field limits of kernels in the context of statistical learning theory, so the focus of this work and [15] is rather different. The contributions of this work, in particular in comparison with [15], are as follows
> 1. Extending and completing the theory of mean field limits of kernels and RKHS
> * Theorem 2.3.2: We show that the limiting function (existence has been established in [15, Corollary 4.3]) is actually in the RKHS of the mean field limit kernel, and shares the same RKHS norm bound. On the one hand, this result is necessary for the proofs in Sections 3 and 4. On the other hand, it completes the commutative diagram in Figure 1, which was already suggested in [15], but an important link was missing. Note that the proof of this result  requires considerably more sophisticated tools than those used in [15].
> * $\Gamma$-convergence inequalities from Lemma 2.4, 2.5: These results are necessary for the proofs in Sections 3 and 4, and make the theory of mean field limits of kernels much easier to handle. The idea to use $\Gamma$-convergence in the present situation and establish the necessary inequalities is a contribution of this work. Interestingly, it seems that this is also the first time that $\Gamma$-convergence arguments have been used in the context of reproducing kernels.
> * Theorem 2.3.1: In contrast to the results in [15], we do not need to use a further subsequence. On the one hand, this is necessary for the $\Gamma$-convergence results, on the other hand, it completes the commutative diagram from Figure 1.
> 2. Approximation and mean field limits of kernels
> * Proposition 3.1, Remark 3.2: The approximation capabilities of mean field limit kernels have not been investigated in [15].
> * Theorem 3.3: In many cases, numerical implementations of kernel methods rely on various forms of the representer theorem, and such a result in the mean field limit is completely new. Furthermore, it seems that this is also the first representer theorem that considers the situation of limits of kernels.
> 3. Statistical learning theory and mean field limit kernels
> * Statistical learning theory setup in the mean field limit: The formalization of such a mean field limit, including the existence result in Proposition 2.1 and the notion of mean field convergence of probability distributions, is new (and nontrivial).  Furthermore, it appears that this is also the first instance of a limit of statistical learning problems that has been investigated.
> * Convergence results for SVMs and their risks (Section 4): All of these results (and their setting) are new. Note that we also need new and more sophisticated techniques compared to [15] to prove these results.
>
> >What is the relationship and difference of this study compared to neural tangent kernel? What is the advantage of mean field kernels?
>
> This is an excellent question. Mean field limit kernels and the neural tangent kernel (NTK) are very different objects, though both arise through a large scale limit (i.e., the number $M$ of entities going to infinity). Some of the technical differences are
> * Mean field limit kernels arise as the limit of kernels, whereas the NTK is a way to connect neural networks to kernel methods
> * The overall input space stays the same for the NTK (the number of hidden units goes to infinity), whereas the input space changes for mean field limit kernels
> * The convergence notions are different (NTK usually involves convergence in probability, we consider mean field convergence results)
> It is difficult to say whether the NTK or mean field limit kernels are "better" , since these objects are very different. Using mean field limit kernels in the context of neural network theory could be interesting, but we have not investigated this question. The main motivation for mean field limit kernels (and their practical strength) is their usage in the context of kinetic theory, as outlined in the introduction (see also the general response above for a more detailed description of our motivation).
>
> >Any concrete examples for mean field kernels?
>
> In this work, we focus on the general theory, which we consider an advantage - all of our results are immediately applicable to any sequence of appropriate kernels with a mean field limit. Large concrete classes of mean field limit kernels (more precisely, kernel sequences that have a mean field limit kernel) have been in introduced in [15], in particular,
> * Kernels from the well-known pull-back construction
> * Double-sum kernels
> Since this work focuses on the theory of mean field limit kernels in statistical learning (which required extending the general theory of these kernels, cf. Section 2, and dealing with approximation questions, cf. Section 3), we decided to not include this aspect here.
>
> Actually, we recently derived an analytical form of a mean field kernel, which we use for numerical experiments in the context of interacting particle systems. However, we found that including this (and the relevant background from kinetic theory) would deteriorate the quality of the submission (since it dilutes the focus).

---

### Official Review · Reviewer_sYSX · 2023-07-07

**Soundness:** 3 good
**Presentation:** 2 fair
**Contribution:** 3 good
**Rating:** 5
**Confidence:** 3

**Summary:**

This paper studies the theory of RKHS consisting of the functions over the space of probability distributions. Complementing the related study [15], the work develops the particle-based approximation theory to RKHS and develops the support vector machines fed distributions as inputs. Moreover, the statistical consistency is also proven.

**Strengths:**

The paper organizes well the theory of RKHS defined over the space of probability distributions. The results seem technically correct. There are potentially many applications in machine learning. Hence, this work can be of interest to the machine learning community.

**Weaknesses:**

- While I recognize many potential applications, it would be nice if the authors could provide concrete examples and datasets to emphasize the importance of the paper.

- The usefulness and computational complexity of the learning setups studied in the paper are somewhat unclear because of the lack of specific examples and experiments.

**Questions:**

Readers might be interested in the particle complexities as $M\to\infty$ because the large number of particles can affect the computational complexity. How large is $M$ needed to achieve a given accuracy in a certain sense?

**Limitations:**

It is somewhat unclear how many applications can exist. It would be nice if some applications were mentioned.

---

> ### Author Rebuttal · Authors · 2023-08-08
>
> Thank you very much for your insightful review. We now address your concerns and answer the posed questions.
>
> **Weaknesses**
> >While I recognize many potential applications, it would be nice if the authors could provide concrete examples and datasets to emphasize the importance of the paper.
>
> We have decided to focus on the theory in order to avoid overloading the submission (presenting the theory properly with all relevant details already takes up a lot of space). Furthermore, we would like to stress that beyond concrete applications, the theory itself appears to be interesting and relevant for the machine learning community.
> * Kernels and RKHSs are important and well-studied, but the type of kernel limits we consider (motivated by concrete applications) is new
> * To the best of our knowledge, the limit of statistical learning problems as introduced and investigated in Section 4 is new (even the notion of such a limit setup)
> * We connect kinetic theory (MFL, $\Gamma$-convergence) with the theory of reproducing kernels
> Finally, regarding concrete examples and datasets, see the paragraph below.
>
> >The usefulness and computational complexity of the learning setups studied in the paper are somewhat unclear because of the lack of specific examples and experiments.
>
> The focus of this submission is on the kernels / RKHSs / statistical learning problems / SVMs in the mean field limit, investigating and discussing computational complexity and practical issues is unfortunately beyond the scope of this submission. This approach is similar to past NeurIPS papers on related topics, for example [12]. We agree that these are relevant issues, but in our opinion, they should be carefully treated in a separate work.
>
> **Questions**
> >Readers might be interested in the particle complexities as $M\rightarrow\infty$ because the large number of particles can affect the computational complexity. How large is $M$ needed to achieve a given accuracy in a certain sense?
>
> This is an excellent and important question. Regarding the computational complexity, when working with the limit objects directly (e.g., an SVM with a MFL kernel), a large $M$ is not a problem -- after all, this is one of the main motivations of this approach. This is completely analogous to the situation of kinetic PDEs, where only the state dimension (corresponding to our dimension $X$), but not the number of particles (here $M$) plays a role.
>
> Regarding the question of accuracy, we indeed do not provide rates of convergence in the mean field limit. To the best of our knowledge, such rates are readily available even in classic applications of kinetic theory, and hence our setup inherits this feature. Borrowing statistical terminology, our results are about consistency, not rates. However, we argue that already the convergence per se (i.e., consistency) is interesting, relevant, and helpful (since it justifies a kinetic theory approach for machine learning methods in this setting and might lead to more efficient numerical methods applicable on the equivalent mesoscopic level). Furthermore, note that having rates w.r.t. $M$ might not be very consequential in practice, since the $M$ is determined by the problem (e.g., the size of the population of interaction particles), and cannot be chosen by the learner (in contrast to the data set size $N$, which often can be increased by simply collecting more data). Nevertheless, providing rates for the mean field limit convergence is a very interesting question that should be investigated in future work.
>
> >It is somewhat unclear how many applications can exist. It would be nice if some applications were mentioned.
>
> Due to space constraints, and our focus on theoretical questions, we have not been able to discuss applications in detail. However, we are motivated by concrete learning problems, as we outlined in the introduction. Since the question of applications/learning scenarios was raised by several reviewers, we have elaborated on this in the general author's answer above.

---

> > ### Comment · Reviewer_sYSX · 2023-08-18
> > **After reading the rebuttal**
> >
> > Thanks for the response. A main concern raised by reviewers was the applicability of the theory to real applications. And the authors have adequately addressed this concern and I have been convinced of its significance. Thus, I would like to raise the score: 4 -> 5.

---

### Official Review · Reviewer_NtYB · 2023-07-09

**Soundness:** 3 good
**Presentation:** 3 good
**Contribution:** 3 good
**Rating:** 5
**Confidence:** 3

**Summary:**

This paper develops mathematically rigorous construction of the mean field limit of kernels of probability measures, which is obtained as a limit of a sequence of kernels with increasing input dimensions, and its application to SVMs.
This is motivated by the analysis of interacting particle systems, when many observations of particles are available and we want to consider a function governing the dynamic as a function of a probability measure rather than a finite-dimensional vector.

Specifically, this paper first confirms the validity of such kernels by extending the existing results: the existence of the mean field limit of functions and kernels, and relations between a sequence of kernels and the mean-field limit of kernels.
Then, they prove the representer theorem (a mean-field limit of a sequence of functions uniformly approximated by each corresponding RKHS functions can also be approximated by the mean-field limit of RKHS functions) and convergence of the minimizers.
Finally, the results are applied to an L2-regularized (with $\lambda \geq 0$) loss minimization problem over RKHS.
They show the existence and uniqueness of the solution (when $\lambda > 0$) and convergence of the loss value.

**Strengths:**

### Problem settings are well-motivated

The purpose of this paper is to develop a foundation of the theory of kernels with probability measures as input in order to view the state of an interacting particle system not as observations of a finite number of particles, but as a function of a probability measure that represents the distribution of the particles. The problem of learning a function with infinite dimensional input may have many other applications, and the potential impact in this area is of a certain significance.

### Rigorous theory for infinite-dimensional input functions

This paper investigates the formulation and thus the basic properties of RKHS for the limit of a sequence of kernels of increasing dimension. However, it should be noted that the main focus is basically on how to define the limit, and not on how to use it specifically. It can be said that this research has a certain novelty if there is no similar discussion.

**Weaknesses:**

### Misleading title

It looks like a paper about generalization error analysis for the community of mean-field neural network optimization. Therefore, I strongly recommend that the authors change the title to characterize their contents more precisely.

### Missing discussion on whether the existing analysis on learning infinite-dimensional input functions can be applicable.

The final part of this paper is essentially a problem of learning functions with infinite-dimensional inputs, but the discussion on the previous works / applications is missed. The authors should make a remark on whether such existing theories are applicable and hopefully conduct some experiments to verify their theory. An example of previous work I think is:

F. Ferraty, A. Mas, and P. Vieu. Nonparametric regression on functional data: inference and practical aspects. Australian & New Zealand Journal of Statistics, 49(3):267–286, 2007.

### The authors say that the first part of the paper is improvement from an under review paper. However, I cannot know the difference and whether such improvement is necessary for the rest part.

This paper consists of the three part. (i) Definitions of functions and kernels in the mean-field limit, (ii) approximation ability and convergence of the minimizers, and (iii) application to SVMs based on (ii).
The authors argue that (i) is improvement from an under review paper. However, I cannot see the difference and whether such improvement is necessary for deriving (ii) and (iii). If not, this paper looks like an assortment of different results. Also from the perspective of preventing double submission, I think the authors should clarify the difference between their results in (i) and whether such an improvement is necessary.

**Questions:**

- Is it possible to add any experiment to the paper?

- Could you tell me more about the difference between Prop. 2.1 and the previous work, and whether such improvement is necessary for the rest part?

**Limitations:**

In order to avoid confusion with optimization and generalization theory of mean-field neural networks, the authors are encouraged to change the title.

---

> ### Author Rebuttal · Authors · 2023-08-08
>
> Thank you very much for your very detailed and careful review. Below we address your concerns and answer the posed questions.
>
> **Weaknesses**
> >Misleading title [...]
>
> Thank you for pointing out this risk of confusion, which we weren't aware of. To avoid any possible confusion, we suggest to change the title to "On kernel-based statistical learning in the mean field limit". We would argue that the term "mean field limit" (MFL) should remain in the title, since this is the accepted term of the considered setting (cf. [7]).
>
> >Missing discussion on whether the existing analysis on learning infinite-dimensional input functions can be applicable.
>
> While functional data analysis (FDA), as considered in Ferraty et al, also deals with infinite-dimensional data, the crucial difference is that the infinite-dimensional data here arises as a particular limit (the MFL) of finite-dimensional data, and we are interested in this limit (and its consequences for kernels / kernel-based methods). Furthermore, the typical data in FDA is rather different ($L^2$ functions, often on $[0,1]$). The situation is similar with kernels for infinite-dimensional inputs: Of course, after going to the limit, we are in this realm (as mentioned in the Introduction, cf. [12]), but this work is about the limits themselves (definition, existence, properties) and the repercussions/chances for kernel methods/theory.
> We will add a corresponding discussion to the Introduction to make this aspect clearer.
>
> >The authors say that the first part of the paper is improvement from an under review paper.
>
> The paper [15] has been accepted and published in the meanwhile (doi: 10.3934/krm.2023010), we will update the bibliography accordingly.
>
> >This paper consists of the three part. [...]
>
> The main contributions in part (i) (Section 2) are
> 1. The MFL function in Theorem 2.2 (existence was already established in [15, Corollary 4.3]) is contained in the MFL kernel RKHS, and shares the same RKHS norm bound.
> 2. Lemma 2.4, 2.5, necessary for the $\Gamma$-convergence arguments used in Sections 3 and 4. These results are novel (in particular, not contained in [15]), and the corresponding proofs are technically demanding and use much more sophisticated arguments than those utilized in [15]. Furthermore, these results are necessary for most of the proofs of the results in parts (ii), (iii) (i.e., Sections 3 and 4).
>
> Additional contributions in part (i) are
> 1. Proposition 2.1, which is necessary for the statistical learning theory setup in Section 4, cf. the Questions paragraph below.
> 2. Avoid going to a subsequence in Theorem 2.1. On the one hand, this is necessary for the $\Gamma$-convergence arguments used later on, on the other hand, this makes the commutative diagram (Fig 1) and its interpretation nicer.
>
> We would like to emphasize that this submission is building on [15], but nothing claimed as a contribution of this submission is contained already in [15].
>
> **Questions**
> >Is it possible to add any experiment to the paper?
>
> We decided not to do so in order to keep the manuscript focused on the theoretical aspects (which take up already a significant amount of space). However, we are working on numerical experiments and empirical evaluations of the concepts introduced here. In our opinion, including such experiments (and the required background on interacting particle systems, kinetic equations, Monte Carlo methods, and numerical methods for kinetic PDEs) would severely degrade the quality of the presentation and make the submission less readable.
>
> >Could you tell me more about the difference between Prop. 2.1 and the previous work, and whether such improvement is necessary for the rest part?
>
> Proposition 2.1 is a generalization of a well-known result (cf. [6, Lemma 1.2]). More precisely, Prop 2.1 allows sequences of functions with a non-compact input domain, and existing results like [6, Lemma 1.2] only work for compact input domains.
> This extension is necessary because in general the third argument of the loss functions in Section 4 cannot be restricted to a compact set. Typically, a loss function $\ell_M$ is used as $\ell_M(\vec x_M, y, f_M(\vec x_M))$ for some input $\vec x_M$, output $y$, and hypothesis $f_M\in H_M$ (using the notation from the submission). However, in the regularized optimization approaches over $H_M$ as considered here, like the minimization problem
> $$
> \min_{f_M\in H_M} \mathcal{R}_{\ell_M,D_N^{[M]}}(f_M) + \|f_M\|_M^2,
> $$
> all $f_M\in H_M$ are allowed. This means that in $\ell_M(\vec x_M, y, f_M(\vec x_M))$, any $f_M\in H_M$ could appear, and since even for a fixed $\vec x_M\in X^M$ we hence cannot restrict the range of $f_M(\vec x_M)$ a priori, we cannot restrict the third argument of the loss functions to a compact subset of the real numbers. Note that clipping techniques frequently employed in kernel-based statistical learning theory (cf. [26, Def. 2.22]) unfortunately do not work here. Since we need the MFL of the loss functions $\ell_M$ for our setup, and since existing MFL results like [6, Lemma 1.2] consider only functions with compact input sets, we needed Prop 2.1.

---

> > ### Comment · Reviewer_NtYB · 2023-08-21
> >
> > Dear authors,
> >
> > Thanks for the clarification. I consider the paper technically solid. I decided to hold my score.
> >
> > Best regards,

---

### Official Review · Reviewer_SuzB · 2023-07-25

**Soundness:** 4 excellent
**Presentation:** 3 good
**Contribution:** 3 good
**Rating:** 6
**Confidence:** 4

**Summary:**

This work derives mean-field limits of kernels and their associated Reproducing Kernel Hilbert Spaces. In particular, the authors quantify the relationship between the finite-input RKHS and the RKHS of the mean-field kernel, derive a Representer Theorem for mean-field kernels, and provide asymptotic convergence guarantees for mean-field kernel SVMs.

**Strengths:**

The results are interesting and well-presented. The authors present a complete treatment of the mean-field limit for kernel methods. The proofs are clear and well-written.

**Weaknesses:**

1. While the results are interesting on its own, I am a little skeptical about the applicability of these results in existing ML problems. I think this work would greatly benefit from a more detailed discussion on the motivation and applications. On a related note, I also encourage the authors to present concrete examples of mean-field kernels for ease of exposition.

2. The complete absence of rates of convergence to the mean-field limit (even for specific kernel classes) is somewhat unsatisfactory. I would appreciate it if the authors could comment on that.

**Questions:**

See above. Overall, the results are quite interesting and a bit of work on the motivation and overall presentation would greatly improve the paper. I would be happy to increase my score if the authors are able to address said concerns.

**Limitations:**

Limitations are adequately discussed.

---

> ### Author Rebuttal · Authors · 2023-08-08
>
> Thank you very much for your review and insightful questions.
>
> **Weaknesses / Questions**
> >1. While the results are interesting on its own, I am a little skeptical about the applicability of these results in existing ML problems. I think this work would greatly benefit from a more detailed discussion on the motivation and applications. On a related note, I also encourage the authors to present concrete examples of mean-field kernels for ease of exposition.
>
> We agree that the motivation and the discussion of practical aspects are rather terse, but we wanted to properly present the theory. In the general response above, we have detailed a concrete learning scenario that motivated the present work, on which we want to apply the theory developed here. We will incorporate this into the Introduction, to make the motivation and application scenario clearer.
>
> Regarding examples of mean field kernels: Several general classes of these objects have been described in [15]. Moreover, recently a concrete analytical form of a mean field kernel has been derived which is used for ongoing numerical investigations. However, we believe that introducing and investigating such a concrete instance would not fit the scope of the present work. Actually, we would argue that the abstraction here is a major strength of the work, since all of our results apply to all mean field kernels as introduced in [15], just as the developments in Section 4 are valid for all sequences of loss functions compatible with Proposition 2.1
>
> >2. The complete absence of rates of convergence to the mean-field limit (even for specific kernel classes) is somewhat unsatisfactory. I would appreciate it if the authors could comment on that.
>
> This is an excellent point. In this work, we focus entirely on convergence per se, and not investigating rates of convergence in the mean field limit. On the one hand, convergence itself is already an interesting and challenging problem, and also helpful from a practical perspective (justifying working on a mesoscopic level). On the other hand, having rates w.r.t. $M$ would be interesting for practitioners (having a rule of thumb when $M$ is large enough to confidently rely on a kinetic approximation) and also in connection with learning rates (e.g., considering the simultaneous limit $M,N\rightarrow\infty$). However, getting such rates is non-trivial and to the best of our knowledge usually not readily available even in classic applications of kinetic theory. We therefore considered this (very interesting) question beyond the scope of the present work. Ensuring rates w.r.t. $M$ is actually subject to ongoing work, and we conjecture that it requires additional regularity conditions on the problem (which would also be contrary to our goal here of being very general).

---

> > ### Comment · Reviewer_SuzB · 2023-08-18
> >
> > I thank the authors for their response.
> >
> > Considering the absence of convergence rates to the mean-field limit (even under additional regularity assumptions), I would like to keep my score.

---

### Official Review · Reviewer_m8Hw · 2023-07-26

**Soundness:** 3 good
**Presentation:** 2 fair
**Contribution:** 2 fair
**Rating:** 5
**Confidence:** 2

**Summary:**

In this work, the authors considered learning problems where the input is a interacting-particles / multi-agent system and  studied the mean-field limit of the kernels, RKHS functions and SVM solutions as the number of particles tends to infinity. The authors showed that, essentially, taking the infinite-particle limit is exchangeable with 1) obtaining the RKHS associated with a kernel, 2) obtaining the solution of a variational problem over the RKHS, and 3) obtaining the "SVM solution" within the RKHS.

**Strengths:**

This work continues the exploration by Fiedler et al. (2023) into the mean-field limit of RKHS when we consider interacting-particles systems with a growing number of particles. Compared to Fiedler et al. (2023), the current work extends the theory by 1) tightening the theoretical result on the commutative relation between taking the infinite-particle limit and obtaining the RKHS (no longer requiring taking another subsequence & controlling the RKHS norm of the limiting function, as reported by the authors on Page 4), and 2) expanding the discussion into the mean-field limit of minimization problems in the RKHS. The proofs of these results appear to involve rigorous techniques from functional analysis and gamma convergence, though I was unable to fully verify their correctness.

**Weaknesses:**

While the infinite-particle limit of the learning problem leads us to interesting theoretical investigations, its practical relevance remains to be seen. Overall, the analysis is carried out at a rather general and abstract level, which is perfectly fine for establishing the theory but makes the motivation harder to interpret by the reader. It is therefore not clear to me whether the work could be of great interest to the NeurIPS audience.

**Questions:**

I don't fully understand why the type of functions considered in Section 4 should be called SVM solutions - there is no support vector involved after all, and instead they seem closer to the solutions to ridge regression / ERM with Tikhonov regularization.

The loss function $l_M$ defined on Line 265, Page 7 is a bit non-standard: it also takes the input vector $\vec{x}$ as an argument. Since it complicates things somewhat, it could be helpful to explain in what scenarios this term is relevant.

---

> ### Author Rebuttal · Authors · 2023-08-08
>
> Thank you very much for your careful and detailed review. Below we address your concerns, answer the posed questions, and provide additional comments and remarks.
>
> **Weaknesses**
> >While the infinite-particle limit of the learning problem leads us to interesting theoretical investigations, its practical relevance remains to be seen. Overall, the analysis is carried out at a rather general and abstract level, which is perfectly fine for establishing the theory but makes the motivation harder to interpret by the reader. It is therefore not clear to me whether the work could be of great interest to the NeurIPS audience.
>
> We fully agree that the present work focuses on theoretical aspects, which might pose a challenge to make the motivation and practical relevance clear. However, we decided to use this approach in order to avoid overloading the submission and keep it focused on the pertinent questions. Machine learning for interacting particle / multiagent systems has become a thriving field in the last years (cf. [21] and [R1]), and the challenges of very large-scale systems (corresponding to our $M\rightarrow\infty$ limit) in this context have been barely tackled so far. We are confident that the theoretical foundations we provide will be helpful in this area. In fact, we are currently working on empirical investigations of mean field limit kernels in the context of interacting particle systems where the computational benefit of treating the mean field case will become apparent. However, including these aspects and the necessary background (from interacting particle systems, kinetic equations, numerical methods for kinetic PDEs) is the beyond the scope of this submission, and we think it would actually deteriorate the quality of presention.
> Furthermore, as briefly mentioned in the Introduction, large-scale limits have become very popular in machine learning in different contexts, and we think that our theoretical results are an interesting contribution to this body of work, providing a different perspective.
>
> **Questions**
> >I don't fully understand why the type of functions considered in Section 4 should be called SVM solutions - there is no support vector involved after all, and instead they seem closer to the solutions to ridge regression / ERM with Tikhonov regularization.
>
> We agree that in the rather general setting that we consider, the solutions $f^\ast_{\mathcal{D}_N,\lambda}$, $f^\ast_{P,\lambda}$ etc., do not necessarily correspond to "classic" SVMs. However, we use the terminology from [26], and  decided to follow their convention, cf. the text immediately after (5.8) and before Theorem 5.3 in this reference. This terminology has also been used in other NeurIPS submissions in the past, e.g., [R2]. However, to avoid confusion, we will add some qualifications to it, i.e., that our infinite-sample / empirical SVM solutions correspond to (Tikhonov) regularized ERM solutions over RKHSs.
>
> >The loss function $\ell_M$ defined on Line 265, Page 7 is a bit non-standard: it also takes the input vector $\vec x$ as an argument. Since it complicates things somewhat, it could be helpful to explain in what scenarios this term is relevant.
>
> We are following the rather general setup of [26, Chp 2]. A common special case is a supervised loss function as defined in [26, Def. 2.7], where the loss function does not depend on the input. In this special case, the setup in Section 4 simplifies considerably, e.g., we do not need to consider the mean field limit of loss functions anymore. However, there are practically relevant loss functions that depend on the inputs. A common example is given by unsupervised loss functions as defined in [26, Def. 2.8], which encompass for instance loss functions used for density level detection, cf. [26, Example 2.9]. Finally, in some applications loss functions with all three arguments appear. For example, in classification problems the cost of misclassification might depend on the input (i.e., there are some instances where misclassification is more costly than in others). By treating the general case (as formalized in [26, Def. 2.1]), all of these cases are handled by our results. We hence decided to use this general form here.
>
> **Strengths**
> Finally, we would like to point out that apart from 1) and 2) mentioned in the review, this submission contains two other substantial contributions.
> * The limiting function alluded to in 1) is actually contained in the mean field limit kernel RKHS (and as a byproduct, we can even control the corresponding RKHS norm). While intuitively clear, showing this is highly nontrivial and has not been established in [15], which allows us to "close the commutative diagram" (Fig 1).
> * In Section 4, we provide an appropriate mean field limit setup of the standard statistical learning theory setting, which allows us to formulate and prove the various mean field limit results using mean field limit kernels. To the best of our knowledge, this has not been done before, and required some additional technical contributions (e.g., Prop 2.1 and Def. (10)). While we only provided an initial investigation of this setup, we expect that interesting follow up work can be conducted here (actually, some of it is already in progress).
>
> [R1] Lu, F., Maggioni, M., & Tang, S. (2021). Learning interaction kernels in heterogeneous systems of agents from multiple trajectories. _The Journal of Machine Learning Research_, _22_(1), 1518-1584.
> [R2] Christmann, Andreas, and Ingo Steinwart. "How SVMs can estimate quantiles and the median." _Advances in neural information processing systems_ 20 (2007).

---

### Author Rebuttal · Authors · 2023-08-08

We thank the reviewers for their careful, detailed reviews and their interesting and insightful questions. We have answered each reviewer separately, and hope to have addressed all of their concerns and questions.

In the general part, we would like to address an aspect that was raised by more than one reviewer, namely the motivation and application scenario of the present work.

First of all, we would like to emphasize that the focus of this submission is on theoretical investigations, and in our opinion, the theory itself (beyond specific applications) is interesting and relevant for the machine learning community. Nevertheless, we were motivated by concrete learning problems (as outlined in the Introduction of the submission), which we consider relevant and interesting, and for which the developments presented here will be helpful in our opinion:

Consider e.g. complex interacting particle systems (IPS) / multiagent systems (MAS), that are difficult to model using first principles. Important examples include (see for instance [22])
* Animal populations (swarms of birds, schools of fish, colonies of microorganisms)
* Human crowds (pedestrian movement, gathering at large events like football games or concerts)
* Vehicular traffic (in particular, traffic jams)
* Markets (where the agents can be human or legal entities)
* Opinion dynamics (inside a society), like in [27]

Frequently the state of such a system/population can be easily measured, e.g., by
* Video recordings or image snapshots of bird swarms/schools of fish, microscopy recordings of microorganism colonies
* Aerial imaging of human crowds (e.g., via quadcopters)
* Polling and social media analysis for opinion dynamics

However, some interesting features of the whole system might be more difficult to measure. For example,
* How a swarm of birds or a school of fish will react to an external stimulus (like an approaching predator), given the current state of the population, e.g., whether there will be a change in the density/spread of the population or change in mean velocity.
* Features of a society in opinion dynamics (average happiness, aggression potential, susceptibility to adversarial interventions), given the current "opinion state"
Measuring such features can be difficult, for example, due to a required intervention. Formally, such a feature is a functional of the current state of the system, and since the state is often easy to measure, it would be useful to have an explicit mapping from state to feature of interest. However, since first principles modeling is unlikely to be successful in the domains considered here, it is promising to learn such a mapping from data (see the Introduction for a formalization). Since this is a standard supervised learning problem (state snapshots as input, noisy measurement of the feature of interest at the state snapshot), it is immediately amenable to kernel methods.

Many of the systems considered above can consist of a large number of agents (thousands of birds in a swarm, millions of microorganisms in a colony), and working on the microscopic or individual level becomes impractical. As done in kinetic theory, it might be beneficial to work on the mesoscopic level, considering only the density of particles/agents on state space. For example, when investigating large swarms of birds, the current density of agents over the state space can be easily measured using commodity imaging equipment, whereas tracking individual birds would require high-resolution imaging and working with very high-dimensional data. Formally, going from the microscopic to the mesoscopic level corresponds to letting the number of particles/agents $M$ tend to infinity, and this can be made rigorous using for example the mean field limit. Based on results and experiences from kinetic theory (e.g., [7]), it is reasonable to assume that our feature of interest (a functional on the state space) has a mean field limit as well (see the Introduction and (1)). Returning to the learning problem, we now have a data set on the mesoscopic level, so the input data (state snapshots) are now  probability distributions over the state space. Using kernel methods, we now lift kernels to  probability distributions over kernel spaces, cf. [12]. Now, all methods and results on the mesoscopic level should be connected to the microsopic level through the mean field limit, and this leads exactly to the problems investigated in this submission.

In practice, we would apply the methods on the mesoscopic level to data originating from the microscopic level (though with large $M$). The various convergence results in Section 4 then ensure that the learning outcome on the mesoscopic level (here, the learned functional of the state) is an approximation to the result on the microscopic level for large $M$. This mimics the situation in kinetic theory, where kinetic PDEs (and associated numerical methods) are used for finite, but large $M$ systems.

We hope that we could make this motivation clearer, and we will adapt the Introduction accordingly.

---

### Decision · Program_Chairs · 2023-09-21

**Decision:**

Accept (poster)

**Comment:**

This paper gives a theoretically rigorous construction of an RKHS on the space of probability measures. They provided a convergence guarantee for an infinite limit of the number of particles approximating each probability measure. This yields a complementary result on existing work.
The work is theoretically rigorous and the reviewers are overall positive on this paper. I recommend acceptance of this paper.

BTW, I encourage the authors to include more concrete examples of real world applications of this approach in the final version.